# CogCoM: A Visual Language Model with Chain-of-Manipulations Reasoning

**Ji Qi**[◇†]**, Ming Ding**[†]**, Weihan Wang**[◇†]**, Yushi Bai**[◇†]**, Qingsong Lv**[†]**, Wenyi Hong**[◇†]
**Bin Xu**[◇*]**, Lei Hou**[◇]**, Juanzi Li**[◇]**, Yuxiao Dong**[◇]**, Jie Tang**[◇*]
[◇]Tsinghua University    [†]Zhipu AI

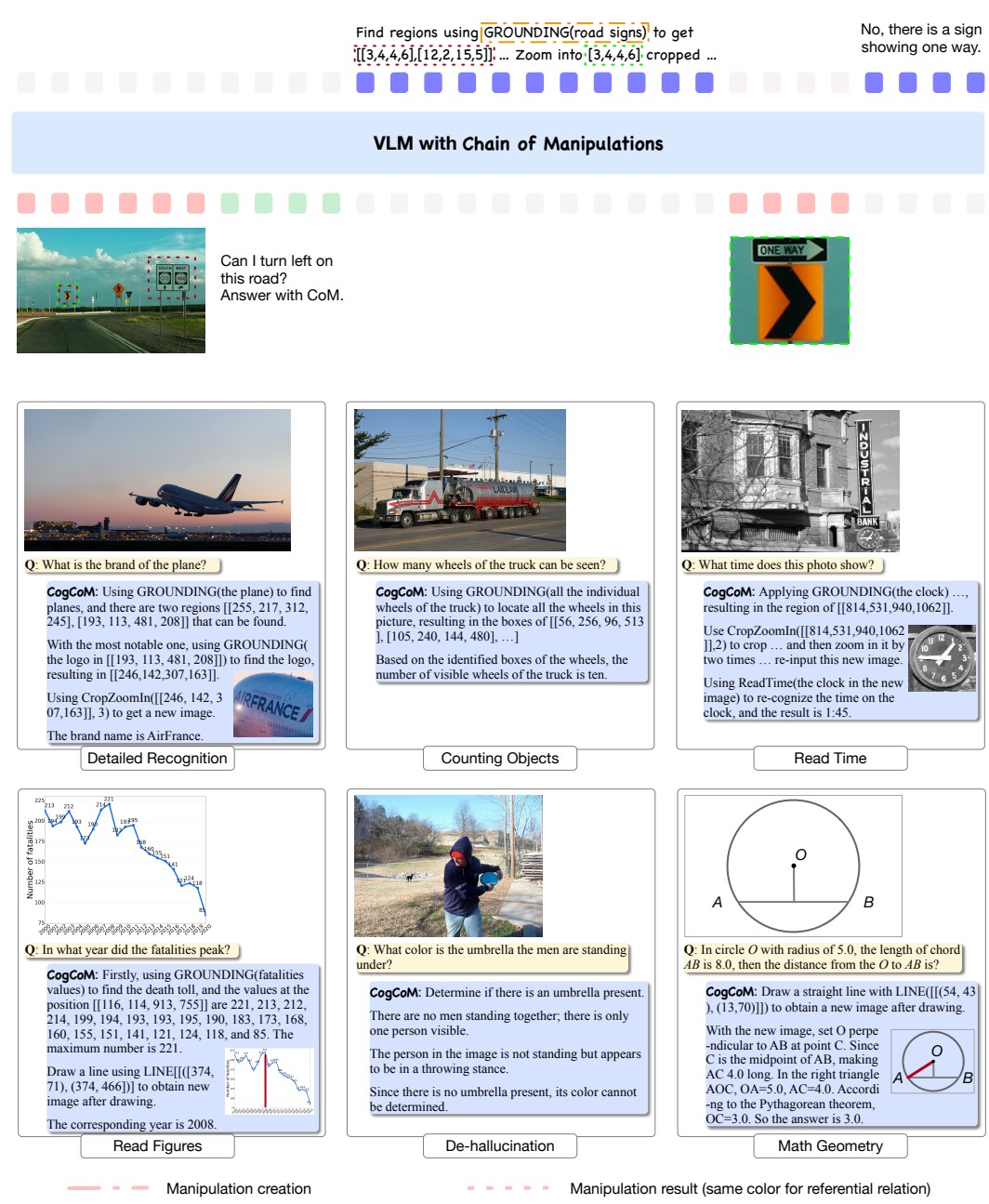

Figure 1: CogCoM solves various visual problems with Chain of Manipulations (CoM) reasoning, which generates evidential and explainable steps **without relying on external tools**.

[†]Work done when JQ, WW, YB, and WH interned at Zhipu AI.
[*]Corresponding authors: BX and JT (xubin | jietang@tsinghua.edu.cn)

## ABSTRACT

Vision-Language Models (VLMs) have shown broad effectiveness due to extensive training that aligns visual inputs with corresponding language responses. However, this conclusive alignment training causes models to overlook essential visual reasoning, leading to failures in handling detailed visual tasks and producing unfaithful responses. Drawing inspiration from human cognition in solving visual problems (*e.g.*, *marking*, *zoom in*), this paper introduces Chain of Manipulations, a mechanism that enables VLMs to tackle problems step-by-step with evidence. After training, models can solve various visual problems by eliciting intrinsic manipulations (*e.g., grounding*, *zoom in*) with results (*e.g.*, *boxes*, *image*) actively without relying on external tools, while also allowing users to trace error causes. In this paper, we study the comprehensive methodology that includes: (1) a flexible design of manipulations based on extensive analysis, (2) an efficient automated data generation pipeline, (3) a compatible VLM architecture capable of multi-turn, multi-image, and (4) a model training process for versatile capabilities. With the design, we also manually annotate **6K** high-quality samples for challenging graphical mathematical problems. Our trained model, **CogCoM**, equipped with this mechanism and 17B parameters, achieves SOTA performance across **9** benchmarks in **4** categories, demonstrating its effectiveness while maintaining interpretability. Code, model, and data are available at `https://github.com/THUDM/CogCoM`.

# 1 INTRODUCTION

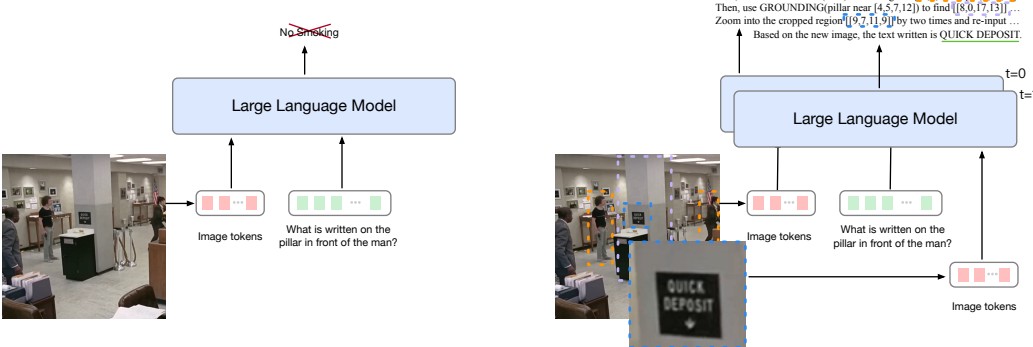

Figure 2: In comparison with existing VLMs, CogCoM performs the multiple steps of evidential reasoning with chain of manipulations (CoM) to achieve the faithful answer to visual scene.

Benefiting from the advantage of Large Language Models (LLMs) in broad world knowledge, large Vision-Language Models (VLMs) (Alayrac et al., 2022; Wang et al., 2023a) that are further trained to understand visual inputs have shown strong capabilities across a wide range of multimodal scenarios, such as visual question answering (Liu et al., 2023b), visual grounding (Peng et al., 2023), and optical character recognition (Zhang et al., 2023b). Research employing VLMs as foundation models (Bai et al., 2023; Sun et al., 2023b; Wang et al., 2023a) typically involves two main training stages, where the first stage develops intrinsic visual understanding through exposure to massive image-caption pairs, while the second stage endows the models with problem-solving abilities via the instruction tuning.

However, existing tuning methods train models to provide conclusive responses to instructions based on visual inputs, causing them to overlook essential intermediate visual reasoning. This often leads to failures in handling visual tasks requiring subtle details, producing unfaithful responses, and even generating hallucinations. For example in the left subplot of Figure 2, we test the top-performing model, CogVLM (Wang et al., 2023a) on the details in the image (*i.e., text written on a pillar*), and it directly responds an incorrect answer (*i.e., NO SMOKING*), most likely due to bias to visual or linguistic priors (*i.e., typical office scenes with a pillar*). The lack of essential visual reasoning about the visual scene can result in a hasty response (Hwang et al., 2023).

Humans solve problems regarding visual details by marking or processing the given images for convenience and precision, which we refer to as manipulations. For example, we identify targets by sequentially locating references, and focus on subtle details by zooming into the corresponding region. Most VLMs develop numerous intrinsic capabilities (*e.g.,* grounding boxes, recognizing text) during the first stage of training. By further imitating the fundamental human behaviours (*e.g.,* cropping, zooming in), models have the potential to perform this cognitive reasoning process. Three major obstacles in enabling VLMs to perform such reasoning are (1) the flexible definitions of manipulations that cover most visual problems, (2) an efficient data collection pipeline capable of producing abundant training data, and (3) a multi-turn multi-image structure that is compatible with existing models.

Inspired by the human cognition in solving visual problems, we introduce Chain of Manipulations (CoM), a mechanism that enables VLMs to solve problems step-by-step with evidence, with each step potentially involving a manipulation on the visual input and its corresponding result, both generated by the model to facilitate the success and fidelity. This paper studies a comprehensive methodology, including **manipulations design**, **data collection**, **model architecture**, and **training process** for developing general VLMs with this mechanism. Based on pilot experiments, we first formally design 6 atomic manipulations, that are capable of addressing diverse visual problems. Next, we propose a cascading data generation pipeline that leverages reliable large language models (LLMs, serving as linguistic annotators) and visual foundational models (VFMs, serving as visual annotators), to automatically generate abundant training data. We collect 70K training samples involving visual reasoning chains using this pipeline. We then devise a multi-turn multi-image model architecture that is compatible with typical VLMs. Based on a data recipe incorporating the curated corpus, we finally train a general VLM equipped with CoM reasoning mechanism, named CogCoM, which possesses capabilities of Chat, Captioning, Grounding and Reasoning. Additionally, benefiting from the expressive capability of the proposed mechanism, we manually annotated **7K high-quality graphical mathematical samples**, each accompanied by a CoM reasoning process, to further advance VLM research in solving challenging mathematical tasks.

We conduct extensive experiments on 9 benchmarks from 4 categories, including TextVQA (Singh et al., 2019), ST-VQA (Biten et al., 2019), TallyVQA (Acharya et al., 2019), and GQA Hudson & Manning (2019) for detailed visual question answering, RefCOCO (Yu et al., 2016), RefCOCO+(Yu et al., 2016), and RefCOCOg (Mao et al., 2016) for visual grounding, POPE (Li et al., 2023d) for hallucination validation, and MM-Vet (Yu et al., 2023b) for general multimodal ability. Our model achieves up to 9.0 and 1.09 accuracy improvement on the detailed VQA and grounding benchmarks, respectively, and the superior performance on the general multimodal benchmark. The results demonstrate the effectiveness of the mechanism while preserving the interpretability of outputs.

## 2 TERMINOLOGY

We first conduct pilot experiments to explore potential atomic manipulations that can address diverse visual problems.

Specifically, given a question about an image, we prompt the advanced large language model, GPT-4, to generate solving steps by optionally utilizing possible actions on the image to facilitate problem-solving. We conduct this experiment on 170K questions from TextVQA, a dataset requiring detailed reasoning and recognition on images. To ensure stability, we manually write 4 demonstrations as priors. The detailed statistics are available in Appendix E.3.

We utilize the StanfordCoreNLP toolkit to extract verb phrases referring to the actions, and the distribution of frequencies is shown in Figure 3. Through result analysis, we find that most of the actions can be mapped to 6 fundamental manipulations on images: *OCR*, *Grounding*, *CropZoomIn*, *Counting*, *Calculate*, and *Line*.

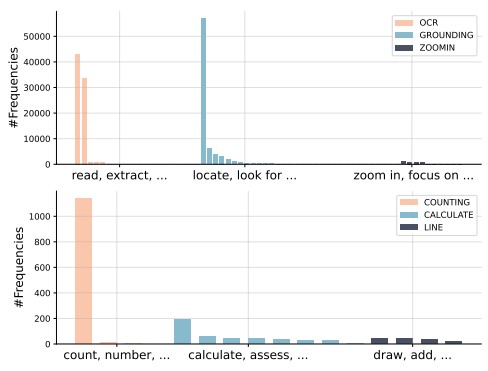

Figure 3: Distribution of the generated 465 actions base on GPT-4, mapped into 6 manipulations.

Based on the observation, we formally predefine a set of 6 atomic manipulations, which can either be developed during pre-training or learned through fine-tuning by imitating human behaviors: $\mathcal{M} \subseteq \{OCR(tgt) \rightarrow txt, Grounding(tgt) \rightarrow bbx, Counting(tgt) \rightarrow num, Calculate(tgt) \rightarrow num, CropZoomIn(bbx, x) \rightarrow img, Line(pts) \rightarrow img\}$, where the corresponding parameters or results $tgt, txt, bbx, num, x, img, pts$ refer to the target description, texts, bounding boxes, numbers, zoom ratio, image, and points, respectively. In addition to the predefined manipulations, we also allow models to create new manipulations during inference to facilitate problem-solving. We empirically find that more complex goals can be derived from these fundamental manipulations.

We then define the **standard CoM data structure** to streamline the subsequent data construction and validation process. Given a question $Q$ about an initial input image $I_0$, a VLM equipped with chain of manipulations mechanism solves the problem to achieve final answer as $VLM_\varsigma(A, C|I_0, Q)$, where $\varsigma$ refers to the reasoning chain with evidence,

$$
\begin{aligned}
\varsigma &= (step_1, step_2, ...) \\
step_i &= (f_i, c_i), \quad f_i \in \mathcal{M}
\end{aligned}
\tag{1}
$$

where $C = (c_i, c_2, ..., c_{|C|})$ refers to the free-form textual descriptions incorporating manipulation names $f_i$ and corresponding results by utilizing $f_i$. This definition explicitly declares the symbolic execution process, while remaining compatible with linguistic reasoning steps. Based on this definition, we can clearly construct standard CoM samples that incorporate the manipulation executions and linguistic steps with evidence. After the data construction, we can utilize a simple method to convert the standard CoM samples to the format of **compatible VQA structure**.

## 3 DATA COLLECTION

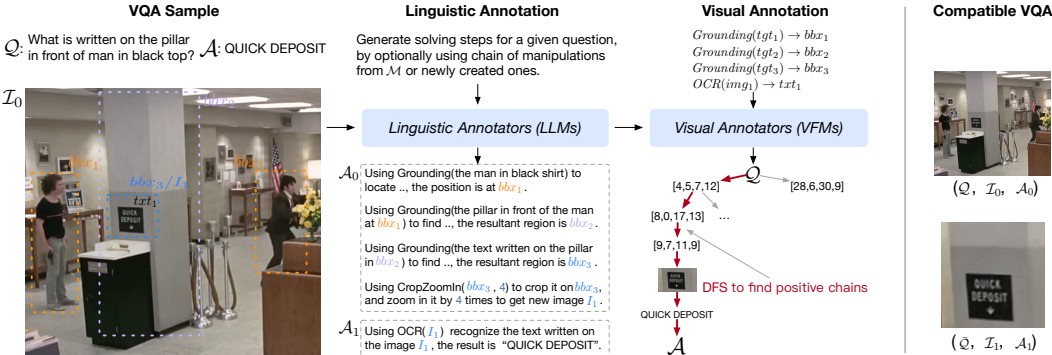

Figure 4: A cascading data generation pipeline that automatically produces standard CoM samples. Given an original VQA sample, the linguistic annotator (LLM) taught with usage of manipulations (prompt) is first asked to provide solving steps for the question $\mathcal{Q}$, and the visual foundational models (VFMs) are then engaged to replace the manipulations results, followed by a final traversal on the tree branched by the possible manipulation results to find positive paths terminating to the answer $\mathcal{A}$.

In this section, we first introduce the automated data generation pipeline (illustrated in Figure 4), that employs reliable LLMs as linguistic annotators and VFMs as the visual annotators to produce CoM samples upon prevalent VQA corpus, and then present the manual annotation of high-quality CoM samples for the challenging graphical mathematical problems.

### 3.1 AUTOMATED DATA GENERATION

Given a general corpus $\mathcal{D} = \{(I, Q, A)\}$ consisting of triplet samples of images with corresponding visual question-answer pairs, our automated data generation pipeline consists of a linguistic annotator and several visual annotators according to the manipulations. For a question $Q$ in each sample, we first engage the linguistic annotator to generate manipulations-assisted solving steps with the CoM format $(f_i, c_i)$, where the corresponding results of the instantiated manipulation executions are set with variables as placeholders. In this paper, we adopt GPT-4 (OpenAI, 2023a), a large language

model with reliable language understanding and generation abilities as the linguistic annotator. We design a comprehensive prompt including the task requirements, usage of manipulations, and output data format, and further manually annotate 5 demonstrations for a stable generation. The detailed implementations are available in Appendix E.4.

We then employ essential visual annotators to supply the results of manipulations requested in the solving steps by exactly performing the corresponding manipulations. By empirically analyzing the manipulations from both predefined set and newly created ones (refers to Appendix E.3 for a detailed statistics), we reveal the *Grounding* and *OCR* are two fundamental manipulations, and most of the others can be consequently derived (*e.g., CropZoomIn* along a region of box, *Counting* upon recognized boxes, and *Calculate* for the recognized formula). Therefore, we employ two visual fundamental models, GroundingDINO (Liu et al., 2023c) and PaddleOCR (Du et al., 2020), and develop the implementations of these manipulations[1]. The execution of the manipulations will transform the sequential reasoning steps into a **tree** $\mathcal{T}$, as the input of current manipulation $f_1(x_a)$ may rely on one of the multiple results of previous manipulations $f_2 \rightarrow (x_b, x_c)$, *i.e.*, $x_a$ rely on $x_b$ (*e.g.*, step 2 for finding pillars in Figure 5). We then perform a Depth First Search (DFS) traversal on each produced tree to find all positive paths $\{\mathcal{P}_i | \mathcal{P}_i \in \mathcal{T}, i = 1, 2, ...\}$, that terminates with the golden answer $A$ as the result of the last manipulation. Based on this recursively searching method, most of the generated positive paths are guaranteed to be error-free. We implement this pipeline on 3 existing datasets that require detailed recognition or counting, TextVQA (Singh et al., 2019), ST-VQA (Biten et al., 2019), and TDIUC (Shrestha et al., 2019), to build 70K CoM samples [2]. The prompt design, an example with linguistic and visual results, and algorithm are available in AppendixE.1.

## 3.2 HUMAN ANNOTATION

The analysis from Fig.1 of AlphaGeometry (Trinh et al., 2024) shows that outputting auxiliary lines in linguistic reasoning process helps LLMs to solve complex geometry problems. Benefiting from the expressive capability of CoM structure, we have also manually annotated high-quality CoM samples for the graphical mathematical problems to facilitate VLMs in solving this challenging scenario. Similar to the automated pipeline, we engage 10 human experts as the linguistic annotators and visual annotators, where each expert is asked to annotate the linguistic solving steps and the use of manipulations, as well as the results of manipulations on images. We perform this annotation on the MathVista (Lu et al., 2023) and ChartQA (Masry et al., 2022), which include geometric and chart math problems, resulting in the collection of 7K high-quality CoM math samples.

Finally, we adapt the CoM samples to be compatible with VQA-style training samples. For each CoM sample including $n$ images with manipulations outputs $(I_0, Q, C_0, I_1, C_1, ..., I_n, A)$, we convert it into a multi-turn VQA sample segmented by the images $[(I_0, Q, C_0), (I_1, \bar{Q}, C_1), ..., (I_n, \bar{Q}, A)]$, where $C_i$ represents the intermediate steps between $I_i$ and $I_{i+1}$, and $\bar{Q}$ is a simple prompt asking model to answer question based on history. This transformation converts CoM samples into multi-turn VQA samples that are compatible with existing VLMs training corpus. The detailed statistics of the data generation are available at Appendix E.3.

## 4 MODEL TRAINING

### 4.1 ARCHITECTURE

We use the same model architecture as CogVLM (Wang et al., 2023a), a general VLM approach that involves four fundamental components: (1) a Visual Encoder, (2) an MLP Adapter, (3) an LLM Backbone, and (4) a Visual Expert Module, for a reliable multimodal understanding. Concretely, the pre-trained EVA2-CLIP-E (Sun et al., 2023a) with 4B parameters and Vicuna-7B-v1.5 (Chiang et al., 2023) are adopted as the visual encoder and LLM backbone, respectively. A two-layer MLP (SwiGLU (Shazeer, 2020)) is further engaged to map the output of the visual encoder into the linguistic space of the LLM backbone. The visual expert module adds the vision-specific weights into the attention layer and feed-forward layer of each block in the LLM backbone, resulting in a total of 6.5B additional parameters for the deep fusion of modalities.

---

[1]We simply implement the *CropZoomIn* referring to human behaviors with a local code interpreter.
[2]The success rate of GPT-4 to achieve the positive paths is 0.3555.

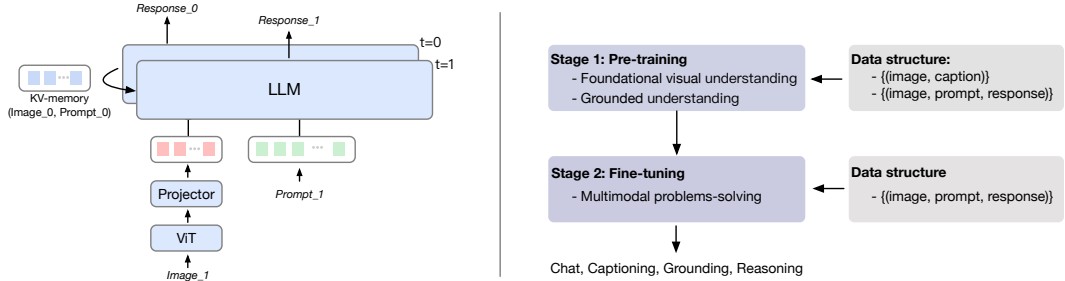

Figure 5: **Left**: A compatible VLM architecture capable of multi-turn multi-image understanding. **Right**: An effective training process to develop a general VLM with versatile capabilities.

Based on this general architecture, we develop a memory-based multi-turn multi-image VLM architecture. Specifically, for a multi-turn VQA sample $[(I_t, Q_t, A_t)|t = 1, 2, ...]$, where $A_t$ refers to $C_t$ in CoM, we keep the accumulated KV memories of each layer in the LLM backbone throughout these turns. And at each turn $t$ in training and inference, we calculate the attention function $att$ as:

$$
\begin{aligned}
att(\boldsymbol{X}) &= softmax(\frac{\boldsymbol{Q}_t \boldsymbol{K}_t'^T}{\sqrt{d}}) \boldsymbol{V}_t' \\
\boldsymbol{K}_t' &= \text{trunc}(\text{concat}(\boldsymbol{K}_0, \boldsymbol{K}_1, ..., \boldsymbol{K}_t)) \\
\boldsymbol{V}_t' &= \text{trunc}(\text{concat}(\boldsymbol{V}_0, \boldsymbol{V}_1, ..., \boldsymbol{V}_t))
\end{aligned}
\tag{2}
$$

where $\boldsymbol{Q}_t \in \mathbb{R}^{s \times d}$ is query representation of current layer, and the $\boldsymbol{K}_t', \boldsymbol{V}_t' \in \mathbb{R}^{(s \times t) \times d}$ refer to the concatenation of accumulated representations and will be further truncated if the sequence length $s \times t$ is greater than a predefined threshold. At $t > 0$, the new image $I_t$ will be cropped from $I_{t-1}$ and amplified with the Bicubic Interpolation (Keys, 1981).

## 4.2 TRAINING

The proposed CogCoM-17B relies on two main stages of training, to develop the capabilities of general multimodal task-solving as well as the visual reasoning.

**First Stage Pre-Training** This stage consists of two ordinal sub-phases of training for foundational visual understanding and grounded generation. Following the pre-training of CogVLM (Wang et al., 2023a), we first train model on 1.5B image-text pairs cleaned from the LAION-2B (Schuhmann et al., 2022) and COYO-700M (Byeon et al., 2022) with 120,000 iterations and batch size of 8,192. We then train model on 40M grounded image-question-answer triples cleaned from LAION-115M (Li et al., 2023b) with 60,000 iterations and batch size of 1,024, where each noun phrase in the answer is followed by a list of coordinates $[[x_0, y_0, x_1, y_1], ...]$[3] referring the phrase to the grounded objects in the image. Both phases adopt the next token prediction objective, and train the 6.5B parameters of visual experts.

**Second Stage Alignment** This stage further trains the model to align with human preferences on solving practical visual problems. We fuse the produced CoM data with 3 types of corpus, including MultiInstruct (Xu et al., 2022), LLaVAR (Zhang et al., 2023b), and ShareGPT4V (Chen et al., 2023b), referring the abilities of instruction-following, texts-recognizing, and detailed-captioning. This fusion results in a total of 570K $(I, Q, A)$ samples, where the answer $A$ in CoM data consists of multiple turns. For the training data of CoM, we randomly prepend a lunching prompt[4] $P^{\mathcal{M}}$ to questions $Q = P^{\mathcal{M}} + Q$ asking models to optionally use manipulations for the adaption of explicitly eliciting. We empirically show that the model can effectively learn the evidential visual reasoning by ingesting this portion of CoM data. We train model with 14,000 iterations and a batch size of 160, where the learning rate reaches $10^{-5}$ after 280 steps of warm-up and then decays linearly. The parameters of 6.5B visual experts are trained with the objective of next token prediction. These two stages of training result in our standard version of CogCoM involving both chat and reasoning capabilities. More training details are available at Appendix F.2.

---

[3] $x_i, y_i \in [000, 999]$ refer to the normalized pixel coordinates.

[4] See Appendix F.1 for examples.

## 5 EXPERIMENT

To quantitatively validate the suitability and efficiency of the proposed method, we conduct experiments on 9 benchmarks corresponding to 4 categories of multimodal capabilities, as well as on a newly constructed testbed that includes the CoM reasoning paths with a keypoints-aware metric. Following previous works, we train two generalist versions of CogCoM to adapt to different scenarios: Visual Question Answering and Visual Grounding, and we evaluate the standard version through a qualitative analysis (Hwang et al., 2023). We also evaluate the time complexity of our model.

- **Detailed Visual Question Answering.** This task involves models to perform detailed reasoning or recognition on images. We use 4 prominent benchmarks including, GQA (Hudson & Manning, 2019), TextVQA (Singh et al., 2019), ST-VQA (Biten et al., 2019), and TallyVQA (Acharya et al., 2019).

- **Visual Grounding.** Visual grounding evaluates the crucial abilities of VLMs in meticulous position understanding. We evaluate our model on 3 standard benchmarks, RefCOCO (Yu et al., 2016), RefCOCO+ (Yu et al., 2016), and RefCOCOg (Mao et al., 2016).

- **General Multimodal Capabilities & Hallucination.** We also evaluate on a general multimodal benchmark, MM-Vet (Yu et al., 2023b), and a hallucination detection benchmark POPE (Li et al., 2023d), to investigate the helpfulness of visual reasoning.

### 5.1 EXPERIMENTS ON DETAILED VQA

VLMs have demonstrated well-known superiority in visual scenes with salient content understanding. We evaluate the effectiveness of CogCoM on VQAs for detailed understanding, which typically require models to perform multiple actions (*find, read*) or multiple reasoning steps (*recognizing and then calculating*). Following previous studies (Wang et al., 2023a), we train our model obtained from the first-phase of stage-1 on a mixture of data, including an instruction corpus of MultiInstruct, 13 publicly available VQA datasets (only using training set), a newly created VQA dataset built through promoting GPT-4V (OpenAI, 2023b) for image-oriented question-answer generation, and the automatically generated 70K CoM corpus. This training results in a generalist VQA model incorporating CoM reasoning. For all existing VQA tasks, we directly prompt CogCoM with given questions and examine the correctness of outputted answers.

| Type | Model | GQA test-balanced | TallyVQA simple | TallyVQA complex | TextVQA test | ST-VQA test |
|------|-------|------------------|-----------------|------------------|--------------|-------------|
| Generalist | Flamingo (Alayrac et al., 2022) | - | - | - | 54.1 | - |
| | GIT (Wang et al., 2022a) | - | - | - | 59.8 | - |
| | GI2 (Wang et al., 2022a) | - | - | - | 67.3 | - |
| | BLIP-2 (Li et al., 2023b) | 44.7$^\dagger$ | - | - | - | 21.7 |
| | InstructBLIP (Dai et al., 2023) | 49.5$^\dagger$ | - | - | - | 50.7$^\dagger$ |
| | Monkey (Li et al., 2024) | 60.7 | - | - | 67.6 | 67.7 |
| | Qwen-VL (Bai et al., 2023) | 59.3 | - | - | 63.8 | - |
| | LLaVA-1.5 (Liu et al., 2023a) | 64.7 | - | - | 62.5 | - |
| | CogVLM (Wang et al., 2023a) | 65.2 | 79.8 | 68.0 | 69.7 | 61.0 |
| | **CogCoM** | **71.7** | **84.0** | **70.1** | **71.1** | **70.0** |
| Specialist SOTAs | | 72.1 (CFR) | 86.0 ( PaLI-X) | 75.6 (PaLI-X) | 71.4 (PaLI-X) | 86.0 (SMoLA) |

Table 1: Performance on VQA benchmarks, where the results with $^\dagger$ refer to the few-shot setting.

### 5.1.1 GQA, TEXTVQA, ST-VQA, TALLYVQA

**Settings** GQA is a compositional VQA benchmark with diverse reasoning questions coming from semantic functional programs. TallyVQA is an object counting benchmark with human-annotated complex counting questions involving challenging non-zero counterparts. TextVQA and ST-VQA are two texts understanding benchmarks requiring models to answer questions through textual cues on images. We use the official evaluation scripts for GQA and TallyVQA, which calculate the accuracy score by the Exact Matching (EM) between model predictions and answers. For TextVQA and ST-VQA, we submit our model predictions to the official online websites for calculating the accuracy with VQA Score metric (Antol et al., 2015).

**Results** As the results shown in Table 2, CogCoM achieves SOTA performance compared to all generalist models, and achieves significant improvements over the baseline model. Specifically, compared to the baseline model, our model achieves up to 5.97 and 9.0 percentage points improvement on the benchmarks that require complex reasoning and detailed recognition, respectively. On GQA and TextVQA, CogCoM also obtains comparable results with the large-scale specialist SOTAs. This result demonstrates the effectiveness of the proposed approach in solving details recognition problem.

### 5.1.2 EXPERIMENTS FOR REASONING ACCURACY AND TIME COMPLEXITY

Due to the lack of resources, we build CoM-test, a benchmark with CoM reasoning chains on the TextVQA test set based on the proposed data generation pipeline, and also introduce a keypoints-aware metric to validate the correctness of reasoning paths (see Appendix E.3 for detailed statistics). We also evaluate the time complexity for model generation on a held-out benchmark, MM-Vet.

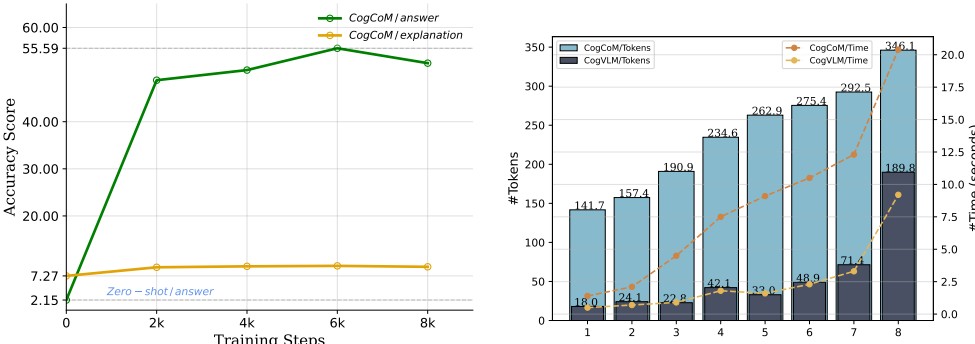

Figure 6: **Left**: Results on a reasoning testbed CoM-test show CogCoM achieves satisfactory performance with only 70K training data and 2K steps. **Right**: Results on MM-Vet show that CogCoM produces comprehensive reasoning content without incurring excessive time overhead.

**Reasoning Accuracy** To validate the correctness of execution and results of manipulations in reasoning paths, we introduce a keypoints-aware evaluation metric that concentrates on these contents and their order. Concretely, given a predicted chain-answer pair $(C', A')$ and the ground truth pair $(C, A)$, we first extract the keypoints (*i.e.,* the name, parameters, and results of manipulations) in $A', A$ to form two lists, and then discretize these two lists into $K'$ and $K$ based on a bag-of-words composed of all keypoints. Then, we calculate the normalized Levenshtein Distance $s_K = Levenshtein(K', K)/N$ as the manipulation score. We also compute the BLEU (Papineni et al., 2002) score $s_C = \text{BLEU}(C', C)$ as the paragraph score. Finally, a weighted average of these two scores serves as the ultimate reasoning score s $acc = (0.6 \times s_K + 0.4 \times s_C)/2$.

We train our first-stage model only using the 70K automated CoM data without other supervision for qualitatively evaluate the effectiveness of chains, and the results are shown in the left subplot of Figure 6. We find that by training with the CoM chains, our model can swiftly achieve the satisfactory performance of 48.41 accuracy score with 2k training steps, and obtain the optimal result of 55.59 with 8K steps. Additionally, the explanation scores gradually improve along with the model performance, indicating that successful reasoning steps contribute to the achieving of final answer.

**Time Complexity** We also evaluate the time complexity and average length of tokens during model reasoning on a held-out test set, MM-Vet. Specifically, we run CogCoM and the baseline model on all 218 questions, and record the time overhead as well as the average number of outputted tokens (using the Vicuna-7B-v1.5 tokenizer). We divide the 218 samples into 8 intervals based on the time expenditure for each sample and calculate the average values of the time complexity and the number of tokens for each interval, with the results presented in the right subplot of Figure 6.

From the results we find that compared to baseline model, CogCoM produces information-intensive reasoning content (*e.g.,* detection boxes, auxiliary lines) without incurring infeasible time overhead. For example, without quantitive optimization, CogCoM outputs 262.9 informative tokens in approximately 9 seconds. With the advantages in long-context optimization techniques (Hooper et al., 2024), we believe that it is crucial for models to produce informative content and accurate responses.

## 5.2 EXPERIMENTS ON VISUAL GROUNDING

The task of visual grounding requires models to precisely provide the corresponding coordinates of regions in an image based on the given target description. Following the existing work (Wang et al., 2023a), we train our model obtained by the first stage on a mixture of datasets, including an instruction corpus MultiInstruct, a high-quality grounded VQA corpus introduced in CogVLM, and the 70K CoM data. This training results in a generalist grounding model that is excelling at visual grounding while capable of reasoning. For all benchmarks, we prompt CogOM in a chat manner to ask the model to provide grounded coordinates, such as "*Where is ⟨expr⟩ answer in [x0,y0,x1,y1] format.*", where the ⟨expr⟩ refers to the target expression. We use the standard metric, that considers a prediction as correct when the intersection-over-union (IoU) between boxes is greater than 0.5.

| Type | Model | RefCOCO | | | RefCOCO+ | | | RefCOCOg | |
|---|---|---|---|---|---|---|---|---|---|
| | | val | test-A | test-B | val | test-A | test-B | val | test |
| Generalist | OFA-L* (Wang et al., 2022b) | 79.96 | 83.67 | 76.39 | 68.29 | 76.00 | 61.75 | 67.57 | 67.58 |
| | Shikra-7B (Chen et al., 2023a) | 87.01 | 90.61 | 80.24 | 81.60 | 87.36 | 72.12 | 82.27 | 82.19 |
| | Shikra-13B (Chen et al., 2023a) | 87.83 | 91.11 | 81.81 | 82.89 | 87.79 | 74.41 | 82.64 | 83.16 |
| | Qwen-VL (Bai et al., 2023) | 89.36 | 92.26 | 85.34 | 83.12 | 88.25 | 77.21 | 85.58 | 85.48 |
| | CogVLM (Wang et al., 2023a) | **92.51** | 93.95 | 88.73 | 87.52 | 91.81 | 81.43 | **89.46** | 90.09 |
| | **CogCoM** | 92.34 | **94.57** | **89.15** | **88.19** | **92.80** | **82.08** | 89.32 | **90.45** |
| Specialist SOTAs | | 92.64 (UNINEXT) | 94.33 (UNINEXT) | 91.46 (UNINEXT) | 88.77 (ONE-PEACE) | 92.21 (ONE-PEACE) | 83.23 (ONE-PEACE) | 89.22 (ONE-PEACE) | 89.37 (UNINEXT-H) |

Table 2: Results on VG benchmarks, where the specialist SOTAs are quoted from (Bai et al., 2023).

**Results**  As shown in Figure 2, CogCoM achieves the best performance in 6 out of all 8 sub-sets. Based on the training with a mixture of broad capabilities, this result indicates that our model exhibits a superior grounding ability while offers potential to solve a variety of tasks. In addition, CogCoM achieves performance on par with the specialist SOTAs and surpasses the ONE-PEACE with a leading performance on the test-A from RefCOCO+. This result demonstrates that under a generalized training integrating multiple capabilities, our model engages grounding as a foundational skill and cultivate the capability to accomplish complex problems.

## 5.3 EXPERIMENTS ON GENERAL MULTIMODAL EVALUATION AND HALLUCINATION EXAMINATION

We further examine the general multimodal capabilities, and the hallucination issue. We use the generalist VQA model and obtain model predictions by directly asking the original questions in benchmarks. We use the challenging adversarial version and official evaluation scripts for POPE.

| Method | LLM | MM-Vet | $POPE_{adv}$ |
|---|---|---|---|
| InstructBLIP (Dai et al., 2023) | Vicuna-13B | 25.6 | 77.3 |
| LLaVA (Liu et al., 2023b) | LLaMA2-7B | 28.1 | 66.3 |
| DreamLLM (Dong et al., 2023) | Vicuna-7B | 35.9 | 76.5 |
| Monkey (Li et al., 2024) | Qwen-7B | 36.2 | - |
| LLaVA-1.5 (Liu et al., 2023a) | Vicuna-13B | 39.4 | 85.0 |
| CogVLM (Wang et al., 2023a) | Vicuna-7B | $45.5^{\dagger}$ | 87.2 |
| **CogCoM** | Vicuna-7B | **46.1** | **87.8** |

Table 3: Evaluation results on the general and hallucination assessment benchmarks.

**Results**  As shown in Table 3, we can see that CogCoM improves the performance by 0.6 points compared to the baseline model on MM-Vet, and achieves the superior performance on POPE which is in consistent with the baseline model. This result suggests that out model maintains superior reasoning capabilities while preserving effectiveness in general multimodal tasks, and simultaneously exhibits lower hallucination.

## 5.4 Ablation Experiment for Training w/wo CoM Data

We conduct experiments on our generalist VQA version model CogCoM-chat, where we removed the 70K CoM training data from the training procedure and keep all other settings unchanged (i.e., using the corpus of public VQAs and 500K training data from MultiInstruct, LLaVAR, and ShareGPT4V). The results on three typical benchmarks are shown in Table4. We can see that our model benefits significantly from the produced CoM training data on these benchmarks that require detailed recognition, multimodal reasoning and mathematical capabilities to achieve substantial improvements.

| Model | TextVQA | MMVet | MathVista |
|---|---|---|---|
| CogCoM (wo) | 64.5 | 45.9 | 34.8 |
| CogCoM (w) | **71.1** ($\uparrow$ 6.6) | **46.1** ($\uparrow$ 0.2) | **35.7** ($\uparrow$ 0.9) |

Table 4: Ablation experimental results for training CogCoM with/without 70K CoM training data.

## 6 Related Works

Most of LVLMs rely on the training on publicly available image-caption pairs, including ALIGN (Jia et al., 2021), MSCOCO (Lin et al., 2014), VG Krishna et al. (2017), CC3M Sharma et al. (2018), CC12M (Changpinyo et al., 2021), SBU (Ordonez et al., 2011), LAION2B (Schuhmann et al., 2022), LAION400M Schuhmann et al. (2021). Starting from Flamingo (Alayrac et al., 2022), a series of LVLMs have focused on training the adaptation layers to align the visual representation to the frozen LLMs on a mixture of image-text pairs with the above corpus, including BLIP2 Li et al. (2023b), KOSMOS Huang et al. (2023b), and OpenFlamingo (Awadalla et al., 2023). Inspired by success of instruction tuning in LLMs (Wang et al., 2022c), a line of works have devoted efforts to build vision-oriented instruction-answer pairs through GPT4 and train models for imitation, such as LLAVA (Liu et al., 2023b), Otter (Li et al., 2023a), VisionLLM (Wang et al., 2023b), MultiInstruct (Xu et al., 2022), Lynx (Zeng et al., 2023), InstructBLIP (Dai et al.), and StableLLaVA (Li et al., 2023c). Recently, researchers have proven the efficiency of developing LVLMs with two stages of training, the first stage of abundant pretraining on image-caption pairs and the second stage of alignment on image-question-answer triples, such as PALI (Chen et al.), PaLI-X (Chen et al., 2023c), Qwen-VL (Bai et al., 2023), and CogVLM Wang et al. (2023a).

To further enhance the ability of LVLMs in solving high-level visual problems, research focusing on various aspects of reasoning is attracting broad attention. We simply divide existing studies into three broad categories. The first line of research focuses on enhance train models with a mastery of cross-modal grounded reasoning, where grounded instruction-following supervision is build through public visual grounding dataset or GPT4-V for training, including KOSMOS-2 (Peng et al., 2023), Shikra (Chen et al., 2023a), and GPT4ROI (Zhang et al., 2023a). The second aspect of efforts have been devoted to promoting models to understand artificial visual scenes, such as figures, charts, and receipts. These studies includes CogAgent (Hong et al., 2023) and CHARTVE (Huang et al., 2023a). Some other studies address the crucial problem of hallucination in LVLMs with counterfactual or interpretable reasoning (Yu et al., 2023a; Yin et al., 2023). V* (Wu & Xie, 2023) also contributes efforts to enhance the details recognition of VLMs based on the LLM-guided searching process.

## 7 Conclusion

This paper studies the problems presented by the conclusive alignment training of VLMs, and proposes a mechanism, Chain of Manipulations (CoM), that enables VLMs to solve problems step-by-step by actively manipulating visual inputs as evidence. We realize this methodology by proposing (1) a flexible data structure, (2) an efficient data generation framework capable of producing abundant samples, (3) a memory-based architecture compatible with existing VLMs, and (4) a training process for versatile capabilities. We also annotate 7K graphical math samples with reasoning chains to facilitate the advancement of VLMs in solving mathematical problems. Experiments on 9 public benchmarks show that our trained 17B general VLM can produce informative reasoning content while achieving superior performance on diverse multimodal problems.

ACKNOWLEDGMENTS

This work is supported by National Natural Science Foundation of China (No. 62277033), and National Natural Science Foundation of China (No. 62476150). Thanks the support from National Engineering Laboratory for Cyberlearning and Intelligent Technology, Beijing Key Lab of Networked Multimedia, and Zhipu AI. This work is also supported by the Natural Science Foundation of China (NSFC) for Distinguished Young Scholar 62425601, and the Natural Science Foundation of China (NSFC) 62495063.

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

## A    DISCUSSION WITH CLOSELY RELATED WORKS

The efforts of **ViperGPT** (Surís et al., 2023) share the same basic idea with our work that decomposes complex visual problems into reasoning steps. In comparison with their training-free framework combining external VLMs using a code LLM, we focus on training an end-to-end VLM to enhance its visual reasoning ability to solve complex problems. **V\*** (Wu & Xie, 2023) is a concurrent work who also solves problems by progressively acquiring visual cues. Their two-part framework first utilizes a VQALLM model to list visual objects, followed by a dedicated search model to acquire the objects. On the other hand, our approach focuses on using one model to actively perform reasoning and to identify or mark the most useful visual information, which may offer the potential for solving more complex reasoning tasks in addition to the detailed identification, such as the challenging geometric math problems. **DualFocus** (Cao et al., 2024) was released around the same time as ours. They also construct a training dataset that includes intermediate cues (bounding boxes) and trained the model with two stages. Compared to their work, our CoM training places more emphasis on answering questions in a single reasoning process and marking images to assist in solving complex problems. **VPD** (Hu et al., 2024) converts programs obtained from LLM and execution engine into CoT and distills them into the VLM, enabling the model to reason when solving visual problems. This approach is similar to our method on visual reasoning. However, our ultimate goal is to train VLMs to solve complex visual problems by actively reasoning (i.e., CoT) and manipulating (e.g., zooming in or drawing auxiliary lines) images which is consistent with the human behavior in realistic scenarios. **VisProg** (Gupta & Kembhavi, 2023) relies on in-context learning of LLMs to generate programs, which are then executed to get final answers. **Ferret-v2** (Zhang et al., 2024) add the Dense Referring and Dense Detection tasks into the training stages and adopts an additional DINO encoder to enhance the fundamental visual capability of VLMs. The study from (Bhattacharyya et al.) trains a VLM to combine the low-level visual features with high-level inferences to reason and generate the final response on videos, where the low-level features are introduced by surrogate tasks during training.

## B    DISCUSSION FOR FUTURE WORKS

Humans solve difficult questions with a period of thinking before answering. In contrast, LLMs/VLMs generate outputs immediately after prompting tokens. The Chain of Thought (CoT) mechanism serves as an effective substitute for this thinking process. Similar to human reasoning, a backtracking mechanism could provide a reliable approach for forming correct and concise reasoning paths. However, as the integration of backtracking results in a high time complexity, developing an efficient backtracking strategy that improves the accuracy of reasoning paths without sacrificing performance is a crucial direction for future work.

## C    LIMITATION AND IMPACT

Though we build a robust framework with remarkable LLMs and reliable visual tools, there are still limitations. First, the diversity of solving steps is still insufficient, and inaccuracies in visual tools (e.g., coarse grounding boxes, OCR errors on slanted text) result in many negative paths, which could be better utilized. We propose improving these issues with dedicated prompts and enhanced visual tools. Second, re-inputting manipulated images with fixed prompts can slow down the process. This can be optimized by incorporating physical manipulations directly into the vector space calculations.   This work presents a general visual reasoning mechanism that alleviates the problems caused by existing conclusion-alignment training for VLMs, introduces a data production framework involving LLMs and visual tools as reliable annotators, and devises a memory-based compatible VLM architecture. We expect this work to bring three benefits to the community. First, the proposed visual reasoning mechanism may push the progress of VLMs in solving complex visual problems. Second, the introduced data production framework may be applied to widespread training scenarios to promote the development of current data-driven machine learning. Third, we hope that the memory-based architecture will be helpful for VLMs in multi-turn long contexts.

# D  ADDITIONAL ANALYSIS

## D.1  THE NUMBER OF INFERENCE TURNS AND THE QUALITY CONTROL

We have detailed the statistics for the training and testing data in the Appendix E.3, including the total number of reasoning chains, the average number of reasoning steps, and the average number of manipulation types. To clarify the average and maximum numbers of the multi-image inference turns, we have also conducted statistics on the number of turns divided by multiple images. The results are as follows: in the training data source from TextVQA and ST-VQA which may involve generating new images such as through zooming, the average number of turns is **1.42**, and the maximum number of turns is **7** (we restrict the maximum number of turns to 4 during training to prevent OOM). In the test set of TextVQA, our model produced an average of **1.54** turns involving multiple images. It is worth noting that not every image requires manipulations such as zooming, and some can be answered through reasoning with evidence or direct observation.

During the collection of our 70K CoM training data, we discard the wrongly recognized data (*i.e.,* we refer to these data as the negative paths in our paper), as these data can not terminate to the golden answer node during the DFS traversal. We used this filtering strategy to ensure that only the correct data capable of reaching the golden answer (i.e., positive paths) was included in the 70K training data. Since we recursively search for the answer by following the intermediate path composed of grounding boxes until we reach the leaf answer node, this approach generally produces the correct path, except in a few cases where the grounding boxes may be too large. Most of the data we have constructed relies on grouping results as intermediate reasoning steps. After our manual verification, we found that most of the reasoning paths are indeed correct. Therefore, most of the generated CoM training samples can be guaranteed to be **error-free**. As using Reinforcement Learning to penalize the negative paths during training is another optimization strategy, we look forward to utilizing these negative paths as negative rewards in future work.

## D.2  THE DISTRIBUTION OF THE SUCCESSFUL AND UNSUCCESSFUL PATHS

Since most of the questions in the constructed dataset ask about the details of images, we have compiled the distribution of image classes for the reasoning data sourced from TextVQA (which includes an image classes label). The statistics are now presented in Figure 7 of the revised paper. Specifically, we analyze the distribution of image classes where a successful reasoning path was not obtained using GPT-4 and visual tools, and among those classes, the distribution of image classes where a successful reasoning path was achieved. To facilitate display, we show the top 200 image classes. To improve visualization, we applied $\log_{10}(\cdot)$ to the positive frequency values on the y-axis, excluding zero. We find that: (1) For most image classes, the proportions of successful and unsuccessful paths are approximately consistent, indicating that the image category does not have a significant impact in this problem. (2) The success rate for constructing reasoning paths is higher for common and prominent objects, such as "person" and "vehicle", while the success rate is lower for smaller, less common objects like "necklace" and "kettle." This suggests that the lower success rate for such objects is due to the limitations in the accuracy of the tools used to find the positive path.

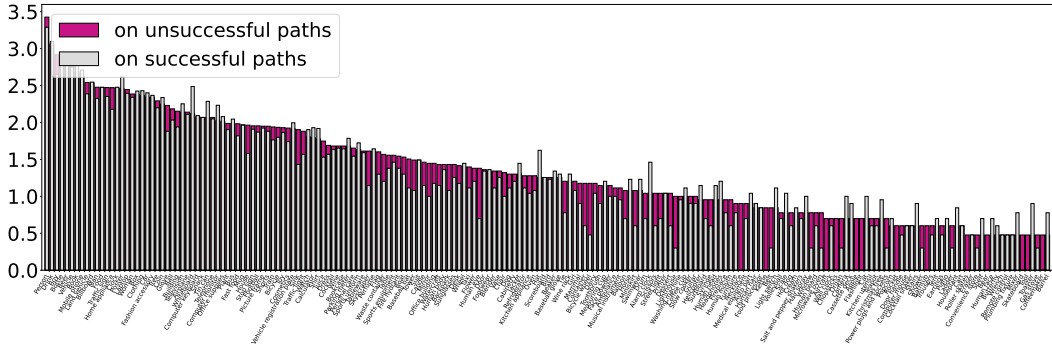

Figure 7: Distributions based on top200 image classes from unsuccessful paths, where the positive frequencies on yaxis are taken as $\log_{10}(\cdot)$ to facilitate the visualization.

# E    DETAILS OF DATA PRODUCTION

In this section, we further introduce the details of CoM data production, with the overall algorithm of a pseudo code, an example of the solving steps generation with LLM and corresponding guidelines, an example of the reasoning chains completion with visual tools. We also list the details of data statistics for the synthesized training data as well as the evaluation data of CoM-test, followed by a limitation analysis for the current data production method.

## E.1    ALGORITHM FOR THE AUTOMATE DATA GENERATION PIPELINE

We provide the pseudocode of the CoM synthesis algorithm to clearly explain the process of data generation, thereby facilitating understanding and reproduction 1.

---

**Algorithm 1** Synthesising Chain of Manipulations

---

1: **Define:** $\begin{cases} Manipulations : \{f_i : x \to y \mid f_i \in \mathcal{M}\} \\ Linguistic\ Annotator : \Psi_L \quad //We\ use\ GPT4\ in\ this\ work \\ Visual\ Annotator : \Psi_V \quad //We\ use\ PaddleOCR\ and\ GroundingDINO\ in\ this\ work \end{cases}$

2: **Input:** Image $I$, Question $Q$, Answer $A$

3: // Linguistic Annotation

4: Prompt $\Psi_L$ with guideline $P^L$ to generate reasoning steps:

$$\varsigma = \Psi_L(Q|P^L), \quad where \begin{cases} \varsigma = (steps_1, steps_2, ...) \\ steps_i = (f_i, desc_i) \end{cases} \tag{3}$$

5: Define tree $\mathcal{T}$

6: **for** $i = 1$ **to** $|\varsigma|$ **do**

7:     Extract $x_i, y_i$ instantiated with $f_i$ in $step_i$

8:     Extract referential boxes $B$ from $x_i$

9:     **for** $b$ in $B$ **do**

10:         Leverage $\Psi_V$ to acquire corresponding visual content $y_i' = \Psi(x_i|I, b)$, and apply $y_i$ to tree

$$\mathcal{T}.level[i].append(y_i) \tag{4}$$

11:     **end for**

12: **end for**

13: Traverse $\mathcal{T}$ to obtain positive chains that leads to given answer with terminal return

$$[\varsigma_1, \varsigma_2, ...] = DFS(\mathcal{T}|A) \tag{5}$$

14: Return $[\varsigma_1, \varsigma_2, ...]$

---

## E.2    THE CoM-TEST BENCHMARK AND EVALUATION METRIC

To measure the correctness of CoM chains, we introduce a **keypoints-aware metric**. The intuition is that we care about the key elements including actions (*i.e.,* manipulation name), targets (*i.e.,* manipulation input), and visual contents (*i.e.,* manipulation returns) of each step in the path, as well as the logical execution order of manipulations. Given a pair of chain-answer annotation $(c, a)$ and corresponding model prediction $(c', a')$, we first sequentially extract the key elements from $c$ and $c'$ to construct two ordered lists, and then replace the elements in the lists with their fixed indices in a Bag-of-Elements $\mathcal{E} = c \cup c'$ to result in lists of $k$ and $k'$. We thus calculate the score as the normalized Levenshtein Distance $s_c = Levenshtein(k, k')/N$ between the two lists, where $N$ is the maximum length between $k$ and $k'$. We adopt this simple discretization strategy with low time complexity to concentrate on the key points as well as the solving order. We further consider the linguistic matching of paragraphs by calculating the BLEU (Papineni et al., 2002) score between two chains $s_p = \text{BLEU}(c, c')$, and the final score is a weighted combination as $acc = (0.6 \times s_c + 0.4 \times s_p)/2$.

### E.3 DATA STATISTICS

We develop a strategy to extract predicate phrases based constituency parsing with StandordCoreNLP, in which we extract verb, conjunction-connected verb phrase, preposition-connected verb phrases.

Besides the standard CoM data incorporating manipulations with explicit visual evidence, the proposed data synthesizing framework is compatible of producing implicit visual reasoning steps $step'_i = (desc_i)$ without involving the manipulations. We thereby also build this partial CoM data on the corpus consisting of absurd visual questions (*i.e.,* asking unanswerable questions based on the given image) to further resist the toxic hallucinations. Specifically, given an image $I$ with a question $Q$, we prompt GPT-4V (OpenAI, 2023b) to solve the question step-by-step to acquire the reasoning chains.

| Data Source | #QAs | #Chains | #Steps/Chain | #Manipulations Types/Chain |
|---|---|---|---|---|
| TextVQA (Biten et al., 2019) | 10782 | 13766 | 2.93 | 2.41 |
| ST-VQA (Singh et al., 2019) | 4814 | 3959 | 2.88 | 2.43 |
| TDIUC-count (Shrestha et al., 2019) | 53547 | 54523 | 2.35 | 0.74 |
| TDIUC-absurd (Shrestha et al., 2019) | 11677 | 11677 | 4.09 | - |
| CoM-test | 4609 | 8612 | 3.26 | 2.18 |

Table 5: Detailed statistics the the training data and evaluation data synthesised with CoM production.

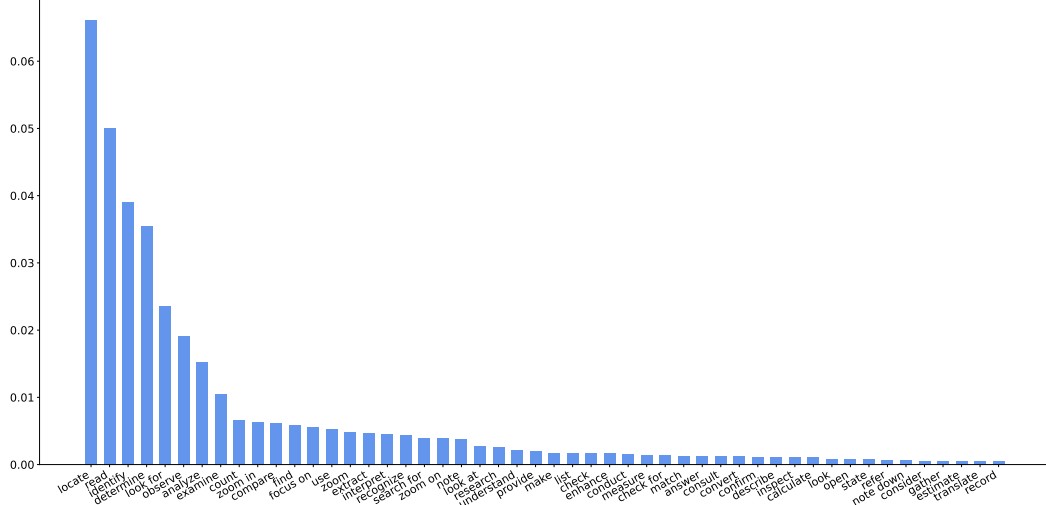

Figure 8: Distribution of the top-50 generated manipulations out of total $465$ based on 4-shot prompting, where the *first three bars* are scaled with $20\%$ for a smooth visualization of all data.

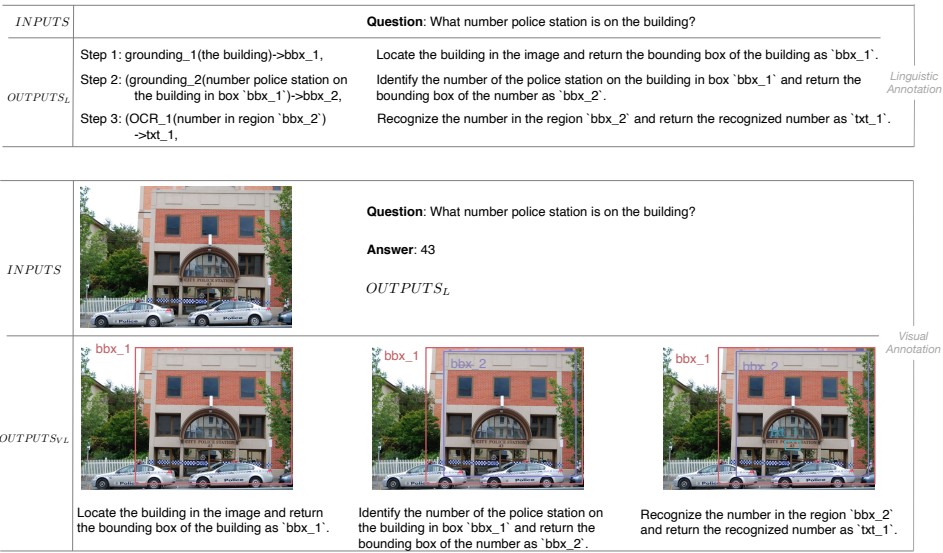

| MANIPULATIONS | OCR_i(tgt)->txt_i: | *i-th OCR manipulation, that recognize the natural texts written on the target `tgt`, and return the recognized texts `txt_i`.* |
| | *calculate(tgt)->res_i:* | *i-th calculate manipulation, that calculate the formula specified by the target `tgt` in current image, and return the calculation result `res_i`.* |
| | *grounding_i(tgt)->bbx_i:* | *i-th grounding manipulation, that locates the object(s) specified by the target noun phrase `tgt` in current image, and return the resulting bounding box(es) as `bbx_i` where each box is represented by the top-left and bottom-right coordinates.* |
| | *crop_and_zoomin_i(bbx)->img_i:* | *i-th crop_and_zoomin manipulation which is useful to identify small and subtle objects in image, that first crops the current image using the box `bbx` defined by the top-left and bottom-right coordinates, and then zoom in the cropped image by two times and finally return the resulting image `img_i`.* |
| REQUIREMENTS (PROMPT) | | *You are a visual assistant capable of generating solving steps for image-oriented visual questions. In each step, you can optionally use a manipulation to operate the image, which can be used to acquire specific information from the image or to acquire the processed new image (please be aware that these manipulations will not actually be performed when you generate the solving steps). The manipulation can be one from the predefined ones, or can be a new one you create yourself (should there indeed be a need), where the predefined manipulations with their descriptions are listed below:\n\n{MANIPULATIONS}.\n\nGiven a question Q about an image, please generate a series of essential solving steps, where the output of each step is a tuple consisting of a Manipulation (leave it to None if the current step doesn't involve any manipulation) and a Description: (1) Manipulation f(x)->y, that is the manipulation `f` targeting `x` to obtain specific information or image `y`; (2) Description, which is a sentence describing the current solving step.\n Please adhere the following format: given an input of 'Q: xxx. The essential solving steps are: ', the output should like 'Step 1: (Manipulation, Description); Step 2: (Manipulation, Description); ...'.\n\n There are several examples:\n {DEMONSTRATIONS}\n\n Q: {QUESTION} The essential solving steps are:* |

Manipulations Definition and Linguistic Annotation Guideline

| *INPUTS* | | **Question**: What number police station is on the building? | |
| *OUTPUTS_L* | Step 1: grounding_1(the building)->bbx_1, | Locate the building in the image and return the bounding box of the building as `bbx_1`. | *Linguistic Annotation* |
| | Step 2: (grounding_2(number police station on the building in box `bbx_1`)->bbx_2, | Identify the number of the police station on the building in box `bbx_1` and return the bounding box of the number as `bbx_2`. | |
| | Step 3: (OCR_1(number in region `bbx_2`) ->txt_1, | Recognize the number in the region `bbx_2` and return the recognized number as `txt_1`. | |

| *INPUTS* | | **Question**: What number police station is on the building?

**Answer**: 43

*OUTPUTS_L* | *Visual Annotation* |
| *OUTPUTS_VL* | | Locate the building in the image and return the bounding box of the building as `bbx_1`. / Identify the number of the police station on the building in box `bbx_1` and return the bounding box of the number as `bbx_2`. / Recognize the number in the region `bbx_2` and return the recognized number as `txt_1`. | |

An Example to show the linguistic annotation results and Visual annotation results

Figure 9: An example shows the configuration, inputs, outputs of the linguistic annotation and visual annotation.

### E.4 DETAILS OF THE LINGUISTIC/VISUAL ANNOTATIONS

In this work, we adopt the GPT4-turbo as the linguistic annotator for generating problems-solving steps, and the API call was conducted during the period of 2023.9 - 2023.12. For the visual annotators, we leverage the currently best-performing tools, GroundingDINO and PaddleOCR, to acquire all visual contents requested by the manipulations. For a clear description for the production setting and results, we illustrate the guiding prompt, and an example-based linguistic annotation results as well as the visual annotation results in Figure 9.

### E.5 LIMITATION ANALYSIS FOR THE DATA PRODUCTION

For the implemented data framework, we engage the remarkable LLM to provide basic solving steps, adopt two reliable visual tools (*i.e.,* GroundingDINO and PaddleOCR) to acquire corresponding visual contents, and then perform the traversal to achieve feasible reasoning paths, which ensures the correctness and robustness of data synthesizing. However, we also find that there are three major limitations caused by the employed models and could be improved in future:

- The lack of diversity in linguistic reasoning steps. The 5-shot prompting to the GPT-4 gains a stable solving steps, but it also results in the descriptions for executing manipulations or general thinking are similar. We suggest that this can be addressed by employing diversified prompts or requirements.
- The inaccuracy of visual tools. We find that there are a considerable amount of negative paths caused by the failures of visual tools, such as the rough granularity of bounding boxes and the error recognition of slated letters or long sentences. This issue can be relieved by improving the semantic understanding capabilities of visual tools.

## F  DETAILS OF TRAINING

### F.1  LAUNCHING PROMPTS

- Please solve the problem gradually via a chain of manipulations, where in each step you can selectively adopt one of the following manipulations GROUNDING(a phrase)→boxes, OCR(an image or a region)→texts, CROP_AND_ZOOMIN(a region on given image)→new_image, CALCULATE(a computable target)→numbers, or invent a new manipulation, if that seems helpful. {QUESTION}
- Please tackle a given question in a stepbystep manner. For each step one of the following manipulations (depicted as Name(Input)→Retrun) can be optionally used: GROUNDING(a phrase)→boxes, OCR(an image or a region)→texts, CROP_AND_ZOOMIN(a region on given image)→new_image, CALCULATE(a computable target)→numbers, or develop a new manipulation yourself (if it is indeed required). {QUESTION}
- Please go through the question incrementally with chain of manipulations (optionally use manipulation when needed) such as GROUNDING(a phrase)→boxes, OCR(an image or a region)→texts, CROP_AND_ZOOMIN(a region on given image)→new_image, CAL-CULATE(a computable target)→numbers, and create a new manipulation if necessary. {QUESTION}

### F.2  TRAINING SETTINGS

| Parameters | Stage1-1 | State1-2 | Stage-2 |
|---|---|---|---|
| Hardware Environment | 3,840 A100xdays | 256 A100xdays | 160 A100xdays |
| Objective | next token prediction | next token prediction | next token prediction |
| Images | 1.5B | 40M | 576K |
| Batch size | 8192 | 1024 | 160 |
| Iterations | 120,000 | 60000 | 14000 |
| Optimizer | AdamW | AdamW | AdamW |
| Learning rate | 1e-4 | 1e-5 | 1e-5 |
| Warm up steps | 7200 | 1200 | 280 |
| Trainable weights | 6.5B visual expert | 6.5B visual expert | 6.5B visual expert |

Table 6: Training details of all stages.

## G  DETAILS OF QUALITATIVE ANALYSIS

### G.1  QUALITATIVE ANALYSIS

We investigate the capabilities of CogCoM on scenarios that requires different types of detailed reasoning, including recognizing textual details, reading time, understanding charts and counting objects. The results are shown in Figure 1. The first case demonstrates that CogCoM finds the region corresponding to the plane logo through two steps of grounding and then achieves the answer based on zooming in the cropped region. The second case illustrates the ability of CogCoM in reading time, by locating the device that displays time and then transforming the time into words based on the read_timne manipulation. In the fourth example, CogCoM first identifies all visible truck wheels through grounding, and then leverages the counting manipulation to consolidate the total number.

## G.2   REPRESENTATIVE CASES

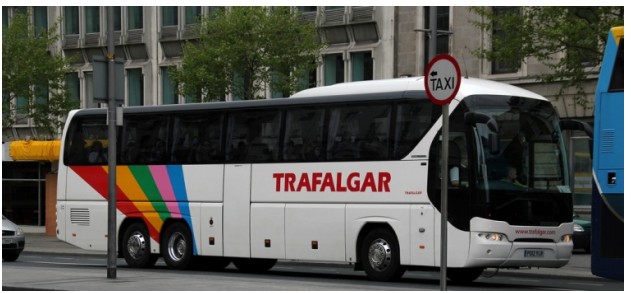

***CoM:***

Given a question, please solve the question step-by-step with a chain of manipulations, where in each step you can optionally use one of the following manipulations (defined as Name(Input)->Return) on the image: GROUNDING(a phrase)->boxes, OCR(an image or a region)->texts, CROP_AND_ZOOMIN(a region on given image)->new_image, CALCULATE(a computable target)->numbers, or a new manipulation created by yourself (if it is indeed needed).

*Question:* What is written on the traffic sign near the bus?
*Model:*   Using GROUNDING(the traffic sign near the bus) to locate the traffic sign near the bus in current image, which is *[[084,157,926,703;676,146,740,273]]*. Utilizing OCR(texts in region *[[084,157,926,703]]*) to read the texts written on the traffic sign within region [[084,157,926,703]], resulting Taxi. So ultimately, the conclusive answer to the question in discussion is Taxi.

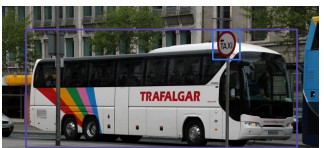

***Visual Grounding:***
   *Question:* Find the region in image that "a traffic sign" describes.
   *Model:*   [[678,145,739,270]

***Grounded Captioning:***
   *Question:* The goal of this task is to generate description for one part of the image. The part is specified by [[678,145,739,270]].
   *Model:*   A no tax sign

***Image Captioning:***
   *Question:* In this task, you are given an image and you will need to generate some text to describe it. Please give me a concise answer.
   *Answer:*   Taxi bus with rainbow colors on side.

***Multi Choice:***
   *Question:* Select the color that does not appear in the rainbow sign of this bus. (1) red. (2) yellow. (3) pink. (4) black. Please give me a direct answer.
   *Answer:*   black

***Detailed Captioning:***
   *Question:* Explain the visual content of the image in great detail.
   *Answer:*   The image captures a scene on a city street, where a white bus with red and blue accents is parked. The bus bears the text "Trafalgar" in bold red letters across its side, indicating it's likely part of a tourist or travel company. The bus is positioned next to a traffic sign that reads "TAXI", suggesting it might be waiting for passengers or ready to embark on a journey. In the background, there are buildings and trees, adding depth to the urban landscape. The overall impression is one of a typical day in an urban setting, with a bus waiting for its passengers amidst the hustle and bustle of city life.

Figure 10: CogCoM demonstrates flexible capabilities for adapting to different multimodal scenarios, including evidential visual reasoning, Visual Grounding, Grounded Captioning, Image Captioning, Multi Choice, and Detailed Captioning.

