# OpenReview forum: "CogCoM: A Visual Language Model with Chain-of-Manipulations Reasoning"
_ICLR.cc/2025/Conference — ICLR 2025 Poster_

### Official Review · Reviewer_qQQK · 2024-10-26

**Soundness:** 3
**Presentation:** 3
**Contribution:** 2
**Rating:** 6
**Confidence:** 4

**Summary:**

The authors provide CogCoM, a technique that augments VLMs with ‘chain-of-manipulation (CoM)’ reasoning. In CoM, given image and text input, VLMs generate visual programs that evoke 6 external modules that can perform OCR, grounding, counting, calculating, zooming-in, and drawing a line. The outputs of external modules are composed into the reasoning path of VLMs. The authors automatically collect 70K training annotations for CogCoM on TextVQA/ST-VQA/TDIUC datasets, by prompting GPT-4 to use GroundingDINO and PaddleOCR, and also collect 7K training annotations for geometry problems with human annotation. The authors compares CogVLM augmented with CogCoM to other VLMs as well as CogVLM in VQA benchmarks (GQA/TallyVQA/TextVQA/ST-VQA), a proposed benchmark for CoM reasoning chain (CoM-test), visual grounding (RefCOCO, RefCOCO+, RefCOCOg), and other general benchmarks (MM-Vet, POPE). CogCoM improves CogVLM performance in many benchmarks, especially on CoM-test.

**Strengths:**

- Introduction of CogCoM, a technique that teaches VLMs to generate visual programs that execute visual expert modules to augment the reasoning chain of VLMs.
- CogCoM improves CogVLM performance in multiple benchmarks.
- Code/Model/Data will be publicly available.

**Weaknesses:**

- **No comparisons with other visual programming methods.** There have been many works where LM or VLM generate programs to execute visual experts for different visual reasoning tasks. Earlier works include VisProg (https://arxiv.org/abs/2211.11559) / ViperGPT (https://arxiv.org/abs/2303.08128). More recent works include GPT4Tools (https://arxiv.org/abs/2305.18752), LLaVA-Plus (https://arxiv.org/abs/2311.05437), Visual Sketchpad (https://arxiv.org/abs/2406.09403) and V* (https://arxiv.org/abs/2312.14135). As generating programs with visual experts itself is not new, qualitative/quantitative comparisons would make the contribution/usefulness of CogCoM more clear.

**Questions:**

- In Fig 2., should the “Large Language Model” be the “Vision-Language Model”?
- In the title, should ‘visual’ be ‘vision’ for consistency?
- Have you also experimented with adding CogCoM to different VLMs? Do you have experiment results showing whether the base VLMs already need to be strong to generate CoM, or can weak VLMs benefit from CoM?

---

> ### Author Response · Authors · 2024-11-21
>
> Dear Reviewer,
>
> First of all, thank you very much for taking the time to read our paper. We would like to clarify that there may be some misunderstanding regarding the method of our paper:
>
> ### **Our model directly outputs the intermediate reasoning process and results in an end-to-end manner during training and inference, without using any external visual modules (tools).  The visual tools are only used for training data construction.**
>
> During automated data construction, we use LLMs and external visual modules to generate CoT data with visual reasoning processes (e.g., asking "What is the logo on the second red cup on the circular table?" based on LLMs to get the solution process: first grounding to find the circular table *A*, then the second red cup *B*, and finally recognizing the logo on the cup). External visual modules are used to obtain manipulation return values such as boxes for *A* and *B* in the CoT data. These synthesized CoT data are used to train our model. After training, the model directly outputs the visual reasoning process and the answer during inference based on its own grounding capability (e.g., output: "I need to first find the circular table which is located at [1,2,8,9], then find the second red cup..."), rather than calling external tools. Given that invoking external visual modules is error-prone and time-consuming, we train the model to think and solve complex problems using this human-like reasoning paradigm, allowing it to generalize to out-of-distribution data.
>
> As we stated in Figure 1 and the Abstract, this is crucial for understanding our work. Below are our detailed responses to your specific questions.

---

> > ### Author Response · Authors · 2024-11-21
> >
> > > ### **Comparisons with Other Visual Programming Works**:
> >
> > Dear Reviewer, thank you for listing the studies related to our work. Due to space limitations, we have included the comparison and detailed discussion of papers closely related to our work in Appendix A — Discussion with Closely Related Works. Please refer to this section for our explanation. We have also listed the background papers on VLMs and VLMs for reasoning in the Related Work section.

---

> > > ### Author Response · Authors · 2024-11-21
> > >
> > > > ### **Should the “Large Language Model” in Fig 2 be the “Vision-Language Model”?**
> > >
> > > **Sorry, the answer is no**. In the model architecture shown in Figure 2, the large language model is part of the overall VLM architecture and performs reasoning by receiving the visual representations outputted by the visual encoder.

---

> > > > ### Author Response · Authors · 2024-11-21
> > > >
> > > > > ### **Should "visual" be "vision" in the title for consistency?**
> > > >
> > > > Thank you once again for your suggestion. When deciding on the title of the paper, we referred to the Flamingo model [1], a typical VLM for visual screen understanding, and used the word "visual" as we believe it is unambiguous. However, we will take your feedback into account and consider adjusting the title properly.
> > > >
> > > > --------------
> > > > [1] Alayrac, Jean-Baptiste, et al. "Flamingo: a visual language model for few-shot learning." Advances in neural information processing systems 35 (2022): 23716-23736.

---

> > > > > ### Author Response · Authors · 2024-11-21
> > > > >
> > > > > > ### **Whether Weak VLMs Benefit from CoM?**
> > > > >
> > > > > Thank you for your insightful suggestion. In this work, our motivation was to address the fact that even VLMs with strong foundational capabilities (e.g., grounding, OCR) are still unable to solve complex visual problems in the same way humans do—through reasoning. Therefore, we trained the model to answer questions by outputting the visual reasoning process, leveraging its own basic visual capabilities (e.g., grounding and OCR).  We tried training on smaller-scale base models and found that VLMs with weaker foundational visual abilities did not see significant improvements in reasoning. Indeed, we believe that for the weak VLMs, the focus should be on improving basic capabilities for solving fundamental tasks. Once these basic abilities are enhanced, the combination of these abilities can be considered for more complex visual reasoning and problem-solving.

---

> ### Author Response · Authors · 2024-11-25
> **Looking forward to your response**
>
> Dear Reviewer qQQK,
>
> Thank you very much for taking your valuable time to review our paper. We have responded to each of your questions individually, and we hope our replies have addressed your concerns. We warmly look forward to hearing back from you.

---

> > ### Comment · Reviewer_qQQK · 2024-11-25
> >
> > Thanks for clarifying the methodology and adding more related works. I have increased the score from 5 to 6 accordingly. I strongly suggest adding more descriptions and analysis of how large VLMs should be to learn the chain of manipulation.

---

> > > ### Author Response · Authors · 2024-11-26
> > >
> > > Dear Reviewer qQQK,
> > >
> > > Thank you very much for your positive feedback. Your recognition means a great deal to us and encourages us to continue making valuable contributions to the  community. We will continuously add descriptions and analysis to demonstrate the effectiveness of VLMs in learning Chain of Manipulations, and propose more effective approaches to further improve the reasoning capabilities of VLMs in our future work.

---

### Official Review · Reviewer_NibN · 2024-11-02

**Soundness:** 3
**Presentation:** 1
**Contribution:** 2
**Rating:** 6
**Confidence:** 4

**Summary:**

This paper presents CogCoM, a model that can reason in a multi-turn way when answering VQA questions. The paper first introduces an approach to collect multi-turn data given only input-output (single-turn) VQA pairs, by using GPT-4, and then uses this multi-turn pseudo-labeled data to finetune a VLM, giving this VLM the capability of reasoning in a multi-turn way. This results in better results in a variety of benchmarks.

**Strengths:**

1 - The paper's contributions are very sound. The idea of collecting data automatically, but having a way to pseudo-verify its correctness (at least end-to-end correctness) is reasonable.

2 - The paper shows good results across a large variety of benchmarks.

3 - The paper additionally contributes extra human-annotated data for mathematical visual reasoning, which can be useful for the community.

**Weaknesses:**

1 - From a scientific point of view, it is hard to tell how significant the contributions are. There is one ablation (which is informative) where they remove the multi-turn training data from their training mix, and it seems like it helps overall (but very little for two out of the three tasks shown). However, it is unclear how much of the contribution comes from the single-turn vs. multi-turn setting, or the data, or the combination of both, or just any other modeling decision when training the final model. If I am not mistaken, the results that are shown from previous methods are all single-turn, which makes the comparisons unfair.

2 - Motivation. There are no comparisons to the module-based approach (ViperGPT-like) that is actually used to collect the pseudo-annotations. If this approach is good to collect the pseudo data, why would it not be used at inference time? Would it perform worse? Is it slower or more expensive?

3 - Presentation

  - There are many grammar and orthography mistakes.
  - The related work is only presented in the appendix. This makes it harder to understand the specific contributions of this paper.

Minor weaknesses:

4 - It is mentioned that the positive paths "are guaranteed to be error free". This is not necessarily true. The model could get to the correct answer through a wrong path, or even by having the last reasoning step completely ignore all previous information.



- Many related literature not mentioned? Or compared
Related work should not go all of it in the appendix!

**Questions:**

- Do the math problems also benefits from the same "modules"? (cropping, detection...), or new ones were identified by human annotators?

- Is there any analysis on the distribution of the correct paths found during the data collection process? If only 35%, do those 35% belong to the same category of questions? Maybe there are entire categories of questions that GPT-4 just never solves, and that end up not being part of the training data.

---

> ### Author Response · Authors · 2024-11-21
>
> Dear Reviewer, we sincerely appreciate your recognition of the reasonable data construction method proposed in our paper, as well as the contribution of the manually annotated data to the community. Your recognition inspires us to engage in more meaningful works. Below are our responses to the questions you have raised:
>
> > ### **The insignificant scientific contribution**:
>
> Beyond a large-scale automated reasoning data construction pipeline and the manually annotated high-quality data upon the proposed design of CoM paradigm, this paper at the first time investigates to incorporate the active visual reasoning on images into a single model through training, which enables the model to possess step-by-step visual reasoning capabilities in and end-to-end manner without relying on any external tools.  Indeed, as we discussed in the Appendix B, how to incorporate the thinking process into LLMs/VLMs is a challenging scientific problem, as current models rely on the transformer-style autoregressive architecture that generate outputs immediately after prompting tokens. Humans solve difficult questions with a period of thinking before answering. The Chain of Thought (CoT) mechanism serves as an effective substitute for this thinking process. Furthermore, how to enable the model to route to different response paradigms (whether to answer directly or output reasoning before answering) is another issue hidden behind the extensive training. Similar to humans’ behavior, we believe that the attempts with extensive results in this paper could bring in a potential for VLMs in solving intellectual and practical scenarios.

---

> > ### Author Response · Authors · 2024-11-21
> >
> > > ### **Unclear Contributions of the Training Data and Inference Paradigm**:
> >
> > - We understand your concerns, but it is precisely because of the design of the CoM reasoning paradigm and the reasoning data collected for this paradigm that the model has learned this reasoning ability. **The two are inseparable**. As shown in Table 4, while both the two models have utilized the *TextVQA training set (i.e., single-turn)*, answering questions based on CoM reasoning (CoM training data + inference-time reasoning paradigm) can effectively improve the model’s performance.
> > - In addition to adding the launching prompt (e.g., Please answer step-by-step with a chain of manipulations.) adapted for CoM reasoning, all models, including ours, use end-to-end inputs and outputs during evaluation. **It is hard to claim that models without reasoning capabilities do not produce reasoning chains with answers, hence making the comparison unfair, because if we were to make a fine distinction, each model employs different design choices and training data.** We tried prepending the prompt "please answer step-by-step" to the baseline model at the time, but this did not lead to any improvement o benchmarks. In fact, this lack of effectiveness is one of the reasons we started this work.

---

> > > ### Author Response · Authors · 2024-11-21
> > >
> > > > ### **Comparisons with Related Works like ViperGPT**:
> > >
> > > Thank you for your suggestion. Due to page limitations, we have placed the comparison and discussion with closely related works in Appendix A - Discussion with Closely Related Works, where we provide a detailed explanation of the differences between our work and others.

---

> > > > ### Author Response · Authors · 2024-11-21
> > > >
> > > > > ### **Why the visual modules were not being used as inference-time execution tools?**
> > > >
> > > > Thank you for your insightful question. In fact, our decision to train an end-to-end model that allows the VLM to output its own reasoning process and results, rather than using external visual modules, is based on two main considerations:
> > > > - **(1) Ineffectiveness of Visual Modules**: As discussed in *Appendix E.5 – Limitation Analysis for Data Production*, visual modules are ineffective in various scenarios (e.g., GroundingDINO struggles with complex semantics, PaddleOCR fails to recognize tilted text). The positive samples collected based on these visual modules from massive data are very limited, with only a 0.3555 success rate. Directly using visual modules would not only be limited by their performance but also result in inefficient reasoning speeds.
> > > > - **(2) Advantages of Large Models**: Compared to external visual modules, VLMs, through training, develop strong semantic understanding and basic visual capabilities. They are also more easily generalizable to distributions outside the training data. For example, when asked a complex semantic question like "What is the position of the second red cup on the circular table?", the trained VLM is able to successfully provide the correct location, while tools like GroundingDINO are unable to answer correctly.
> > > >
> > > > We aim to build reasoning data based on visual modules to enable the large model to learn this reasoning approach and perform the reasoning using its own capabilities.

---

> > > > > ### Author Response · Authors · 2024-11-21
> > > > >
> > > > > > ### **The Presentation Problem**:
> > > > >
> > > > > Thank you very much for pointing out the presentation issues in our paper. We will carefully re-examine the entire manuscript and correct the grammatical and orthographic errors. Due to the workload involved, we will upload the revised version immediately after completing the proofreading. **We have moved the "Related Works" section into the main body of the paper to improve clarity and facilitate better understanding and flow.**

---

> > > > > > ### Author Response · Authors · 2024-11-21
> > > > > >
> > > > > > > ### **The Inaccurate Statement for the Positive Paths**:
> > > > > >
> > > > > > Since we recursively search for the answer by following the intermediate path composed of grounding boxes until we reach the leaf answer node (e.g., for a question like *"What text is on the logo of the second red cup on the circular table?"*, we first identify the circular table *[1,2,8,9]*, then find the second red cup within *[1,2,8,9]* as *[3,4,6,7]*, and finally apply PaddleOCR within the region *[3,4,6,7]* to recognize the text and compare it to the golden answer), this approach generally produces the correct path, except in cases where the grounding boxes may be too large. Most of the data we have constructed relies on grouping results as intermediate reasoning steps. After our manual verification, we found that most of the reasoning paths are indeed correct. However, we agree with your concern about the inaccuracy of the statement. Therefore, we have revised the paper to say "Most of the positive paths can be guaranteed to be error-free," and we have added the above explanation. Please kindly refer to the revised PDF manuscript for details.

---

> ### Author Response · Authors · 2024-11-21
>
> > ### **Regarding the Human Annotation**:
>
> We provided the CoM reasoning approach for solving complex visual problems as a guideline to the annotators, along with several cases to standardize their annotation process. For example, when solving the question in the sixth subplot of Figure 1 for  *"In what year did the fatalities peak?"*, we instructed the annotators to first use the model’s grounding ability to locate the fatalities values in the image. Then, they should draw a vertical line to mark the x-coordinate corresponding to the peak, and finally, identify the year associated with that x-coordinate. **Aside from drawing the auxiliary lines, most of the reasoning operations are similar to those in the automatic data construction pipeline.** In fact, for geometric mathematics problems, we believe that, apart from operations on the graph (such as drawing lines and grounding), most of the reasoning is reflected in the textual thought process and calculations.

---

> > ### Author Response · Authors · 2024-11-21
> >
> > > ### **Statistics for the Successful and Unsuccessful Paths**:
> >
> > Thank you for your valuable suggestions. Since most of the questions in the constructed dataset ask about the details of images, we have compiled the distribution of image classes for the reasoning data sourced from TextVQA (which includes an image classes label). **The statistics are now presented in Figure 7 of the revised paper.**
> >
> > Specifically, we analyze the distribution of image classes where a successful reasoning path was not obtained using GPT-4 and visual tools, and among those classes, the distribution of image classes where a successful reasoning path was achieved. To facilitate display, we show the top 200 image classes. To improve visualization, we applied $\log_{10}(\cdot)$ to the positive frequency values on the y-axis, excluding zero. We find that:
> >  - (1) For most image classes, the proportions of successful and unsuccessful paths are approximately consistent, indicating that the image category does not have a significant impact in this problem.
> >  - (2) The success rate for constructing reasoning paths is higher for common and prominent objects, such as "person" and "vehicle," while the success rate is lower for smaller, less common objects like "necklace" and "kettle." This suggests that the lower success rate for such objects is due to the limitations in the accuracy of the tools used to find the positive path.
> >
> > Based on these observations, the reasonable direction for improvements is to use improved visual tools or manual annotation for small object images and mixed text-image scenarios (such as diagrams in academic papers with accompanying descriptions) to construct more reasoning data and address these challenging multimodal reasoning problems.

---

> ### Author Response · Authors · 2024-11-25
> **Looking forward to discuss**
>
> Dear Reviewer NibN,
>
> We greatly appreciate the time and effort you have dedicated to reviewing our paper and providing such valuable suggestions. We have carefully addressed your questions one by one. We sincerely hope our responses have resolved your concerns and look forward to your further feedback.

---

> > ### Comment · Reviewer_NibN · 2024-11-25
> > **The rebuttal is convincing**
> >
> > I appreciate the detailed rebuttal by the authors. They addressed the points that I raised in detail.
> >
> > I appreciate the explanations on their contribution and why the paradigm and the data are inseparable.
> >
> > I still believe that the paper would benefit from more comparisons with other approaches (both qualitatively and quantitatively). The comparison in the paper simply explains what makes the approaches different, but does not show results, or weaknesses of previous approaches that are solved by the presented approach.
> >
> > Overall, I believe this paper can be useful for the community, so I updated my score to an accept.

---

> > > ### Author Response · Authors · 2024-11-26
> > >
> > > Dear Reviewer,
> > >
> > > Thank you very much for your recognition of our work and your comprehensive suggestions for improvement. Conducting comparisons with works on improving reasoning capabilities of VLMs to explore more effective approaches is indeed crucial. Based on your suggestions, we will thoroughly analyze more recent related works (strengths and weakness) and drawing from these analyses, strive to propose approaches for improving VLM reasoning capabilities in our future work. Additionally, in accordance to your previous suggestions regarding the writing, we have proofread the grammar and spelling throughout the paper and uploaded a revised PDF. We will further revise the paper to ensure the accuracy of the writing.

---

### Official Review · Reviewer_pnHj · 2024-11-03

**Soundness:** 4
**Presentation:** 4
**Contribution:** 3
**Rating:** 6
**Confidence:** 4

**Summary:**

This paper introduces Chain of Manipulations (CoM) to address the reasoning questions that require a detailed look of the visual information. CoM is a step-by-step strategy allowing the model to take manipulation operations like Crop/ZoomIn over the image, thus reserving more capacity for the image regions/bboxes that needs to be looked at with more detail. The contributions include both two datasets and a trained model, which shows SoTA performance on 9 datasets in 4 categories.

**Strengths:**

1. The proposed method, defining operations of image manipulation to solve queries that require a detailed look is intuitive, is intuitive and clear.
2. Intensive experiment results compared with up-to-date models are shown on various tasks, leading to SoTA performance.
3. writing is clear and easy to follow

**Weaknesses:**

1. One thing not very clear - in inference time, once the model is trained, is step-by-step CoM still needed? In other words, is CoM a method for collecting training data, or is it for VLM inference as well? Fig-6 right breaks down the questions into eight groups based on the time overhead - is this time the inference time using one single VLM call or multiple calls using CoM is needed?
2. In principle, how does this work compare to visual programming [1,2]? Is the defined manipulations like ZoomIn/Crop a specific case of the operations as in these works? More discussions will be helpful.
3. While the VQA results show huge improvement, the improvements on grounding on RefCOCO(+/g) are not as significant. Any discussions/analysis on this?
4. The proposed data contains 70k automatically generated data and 7k human-annotated data. How does each of them contribute to the model performance? For example, are there results using each type of the data separately? Or are there more discussions on why both types are needed and only one type is not sufficient?
[1] Gupta, Tanmay, and Aniruddha Kembhavi. "Visual programming: Compositional visual reasoning without training." Proceedings of the IEEE/CVF Conference on Computer Vision and Pattern Recognition. 2023.
[2] Surís, Dídac, Sachit Menon, and Carl Vondrick. "Vipergpt: Visual inference via python execution for reasoning." Proceedings of the IEEE/CVF International Conference on Computer Vision. 2023.

**Questions:**

See weakness.

---

> ### Author Response · Authors · 2024-11-21
>
> Dear Reviewer, we truly appreciate the time and effort you have dedicated to reviewing our paper and offering your insightful comments. Below, we provide our explanations and responses to the questions you have raised.
>
> > ### **Whether the CoM reasoning refers to an inference-time method**:
>
> Yes, the CoM reasoning is an inference-time method. During data collection, we utilize linguistic and visual tools to construct CoM chains (i.e., triples of *<image, reasoning process with visual information, answer>*). At inference, the trained model generates a reasoning process (i.e., outputs the manipulation calls that will be executed by itself with the execution results, for example: first, look for *A* which is at *[x1,y1,x2,y2]*, and then recognize the text in *A* …) and provides an answer to a question without the need for external tools.  Additionally, we randomly add CoM launching prompts during training to preserve the model's autonomy in reasoning decision while also providing the option for explicit launching. At evaluation (e.g., in Fig-6), we prepend the launching prompts to the questions to enable CogCoM to perform reasoning and achieve the final answers.

---

> > ### Author Response · Authors · 2024-11-21
> >
> > > ### **Insignificant Performance on Grounding Benchmarks** :
> >
> > Existing VG benchmarks primarily measure the accuracy of the model in referring phrases to target boxes (also known as referring expression comprehension ). This ability is closely related to the first stage of training (based on 40M grounded image-question-answer triples), which is consistent with the baseline model, CogVLM. CoM reasoning can assist visual grounding in two ways: (1) CoT reasoning problems with intermediate grounding (grounding results appear in the process, e.g., "What is the logo of the second red cup on the table?") and (2) grounding problems with complex semantics (the grounding result is the answer, e.g., "Where is the second red cup on the table?").  CoM reasoning is more effective for grounding scenarios that require reasoning or semantic understanding. TallyVQA can be roughly viewed as a counting problem that requires intermediate grounding results (although it does not evaluate the accuracy of the intermediate process), and CogCoM achieved considerable improvements on this benchmark compared to the baseline model.

---

> ### Author Response · Authors · 2024-11-21
>
> > ###  **How CoM-train and CoM-math Contribute to the Performance Separately**:
>
> The constructed 70K CoM-train data and the manually annotated 7K CoM-math data contribute to the model's capabilities in different aspects. CoM-train is primarily built on **natural images** (which can effectively leverage visual tools like GroundingDINO) to help the model with detailed recognition and logical reasoning. To facilitate the model's ability to solve math reasoning problems on **artificial images** (where visual tools are difficult to apply), we manually annotated the high-quality 7K CoM-math data with intermediate reasoning processes. Both types of data are essential. An empirical finding is that after adding the CoM-math data to the training set, the model's performance on the MathVista benchmark showed a noticeable improvement.
>
> We hope that our clarifications have addressed your concerns. Should you have any further questions, please feel free to let us know.

---

> ### Author Response · Authors · 2024-11-25
> **Looking forward to response**
>
> Dear Reviewer pnHj,
>
> Thank you very much for investing significant effort in reviewing our paper and proving insightful suggestions. We have provided point-by-point responses to your questions. We sincerely hope that our responses address your concerns adequately and we look forward to your feedback.

---

> > ### Comment · Reviewer_pnHj · 2024-11-27
> > **Thanks for the rebuttal**
> >
> > I thank the authors for the rebuttal.
> >
> > The rebuttal addresses weakness 1, 3, 4. I suggest the authors add the numbers for 4, instead of describing that "An empirical finding is that after adding the CoM-math data to the training set, the model's performance on the MathVista benchmark showed a noticeable improvement.".
> >
> > The rebuttal didn't answer Q2, but I do see answer for similar questions from other reviewers (like sBKx). I suggest the authors adding more related work discussion.
> >
> > I will keep my score as 6.

---

> > > ### Author Response · Authors · 2024-12-03
> > > **Response to the benefits of CoM-math in Q2**
> > >
> > > **Dear Reviewer, thank you very much for your reply and insightful suggestions to help us analyze our dataset more thoroughly. In response to your suggestion, we have conducted the following experiment:**
> > >
> > > - we trained our CogCoM-base model for one epoch only using the annotated CoM-math dataset (the loss decreased from an initial value of 1.4 to 0.09), resulting in the trained model CogCoM-base-commath. We then sampled a subset from the MathVista testmini set with 100 examples, which were not part of the annotated dataset, to test both CogCoM-base and CogCoM-base-commath models. The accuracies (extract the predicted answers to match the ground truths) of two models were as follows: **CogCoM-base = 27.1** (zero-shot) and **CogCoM-base-commath = 78.6** (after training). Though comparing the trained model, this result suggest that the annotated CoM-math dataset indeed improve the model's performance on Math-related tasks and brings gains in training.
> > >
> > >
> > > **Additionally, here is a case study for the “images/16.jpg” problem from the MathVista testmini set:**
> > >
> > > - Given the histogram image and the question: “How many items sold less than 5 units in at least one store?”, our model’s reasoning output is as follows:
> > >
> > >   + > “First, by GROUNDING(all items), obtain the sales status of all products; their positions are very[[251,648,370,984]], mill[[403,451,524,984]], cop[[564,224,681,978]], dot[[721,442,842,984]]. Analyze the sales performance of various products; check if there is an item with sales less than 5 in both stores. It is analyzed that the sales volume of ‘very’ products is less than 5 in two stores, the sales volume of ‘mill’ products is less than 5 in one store, and both ‘cop’ and ‘dot’ products have sales volumes exceeding 5 in every store. The sales volume of two products, ‘very’ and ‘mill,’ is less than 5 in at least one store, so the answer is 2. In conclusion, the final answer to this question is 2.”
> > >
> > > The final result derived through this reasoning matches the correct answer, indicating that the model is able to arrive at the correct answer to challenging questions through effective reasoning.

---

### Official Review · Reviewer_sBKx · 2024-11-07

**Soundness:** 3
**Presentation:** 3
**Contribution:** 3
**Rating:** 8
**Confidence:** 4

**Summary:**

The paper focuses on visual reasoning problems using large vision language models. The paper focuses on the problem caused by the first alignment training stage in such vision language models, where the model does not focus on the low level details and instead focuses only on the high level details of the image. The paper proposes to solve this problem by teaching the model to perform intrinsic manipulations (e.g., grounding, zoom in) with results (e.g., boxes, image). The paper shows promising performance on various visual reasoning problems.

**Strengths:**

* The paper shows promising results.
* The paper is well written and easy to understand.
* The paper includes extensive qualitative examples.
* The proposed CoM dataset has the potential to be useful to the broader community.

**Weaknesses:**

* Novelty: The core idea of the paper is very similar to that of "Visual Program Distillation: Distilling Tools and Programmatic Reasoning into Vision-Language Models, CVPR 2024" and "Visual Programming: Compositional Visual Reasoning Without Training, CVPR 2023". The Visual Program Distillation work trains the VLM to perform several grounding tasks for more accurate visual reasoning. Somewhat similarly, "Ferret-v2: An Improved Baseline for Referring and Grounding with Large Language Models, arXiv 2024" uses a visual sampler to extract relevant regions in the image.

* On the other hand, prior work such as "Look, Remember and Reason: Grounded reasoning in videos with language models, ICLR 2024" used surrogate tasks to extract low level visual information. A discussion of the relationship to prior work is necessary.

* Limited size of the CoM dataset: The proposed CoM dataset consist of only 6k images. This might limit its utility due to the lack of visual diversity.

* Ablation studies: The proposed method uses several datasets during the second stage of training along with the proposed CoM data -- this includes MultiInstruct (Xu et al., 2022), LLaVAR (Zhang et al., 2023b), and ShareGPT4V (Chen et al., 2023c). It would be beneficial to include additional ablations to highlight the utility of each of these datasets.

**Questions:**

* The paper should include a broader discussion of prior work.
* The paper should include more extensive ablations.

---

> ### Author Response · Authors · 2024-11-21
>
> Dear Reviewer, thank you for taking your valuable time to review our paper and provide insightful suggestions. Regarding the issues you raised, we have added the following discussions and experimental results. The corresponding changes have been included in the revised PDF manuscript.
>
>
>
> > ### **Discussion of Differences with Previous Works**:
>
> VPD [1] converts programs obtained from LLM and execution engine into CoT and distills these CoTs into the VLM, enabling the model to reason when solving visual problems. This approach is similar to ours in terms of visual reasoning. However, our ultimate goal is to train VLMs to solve complex visual problems by actively reasoning (i.e., CoT) and manipulating images (e.g., zooming in or drawing auxiliary lines), which is consistent with human behavior in realistic scenarios. In experiments, we observed considerable improvements in detail recognition and mathematical problems with essential manipulations. VisProg [2] relies on in-context learning of LLMs to generate programs, which are then executed to obtain final answers. Ferret-v2 [3] incorporates Dense Referring (i.e., allowing the model to recognize all categories of the given boxes) and Dense Detection (i.e., allowing the model to output all boxes of objects in a given image) tasks into the training stages and adopts an additional DINO encoder to enhance the fundamental visual capabilities of VLMs. This is an effective solution to enhance the grounding capabilities with complex semantics of models. The study from [4] trains a VLM to combine low-level visual features with high-level inferences to reason and generate the final response on videos. The low-level features are introduced by surrogate tasks during training, including object recognition, re-identification, and tracking.
>
> Thank you for your valuable suggestion. **We have added these comparisons to our revised paper. Please see Appendix A for details.**
>
> --------------
> [1] Visual Program Distillation: Distilling Tools and Programmatic Reasoning into Vision-Language Models, CVPR 2024.
> [2] Visual Programming: Compositional Visual Reasoning Without Training, CVPR 2023.
> [3] Ferret-v2: An Improved Baseline for Referring and Grounding with Large Language Models, arXiv 11-Apr-2024.
> [4] Look, Remember and Reason: Grounded reasoning in videos with language models, ICLR 2024.

---

> > ### Author Response · Authors · 2024-11-21
> >
> > > ### **Limited size of the CoM dataset:**
> >
> > Thank you for your suggestions for future improvements in our data construction. We have re-checked the statistics for the data we collected. The training set, CoM-train, contains 12,102 images from TextVQA and ST-VQA, as well as 20,699 images from TDIUC. If you are referring to the manually annotated CoM-math data for mathematical reasoning, which includes 4,506 images, we consider these images to be high-quality, labor-intensive data, where each data sample includes a series of reasoning steps from human annotators. However, we acknowledge that the number of images in our dataset is still quite limited. We will continue to add more diverse scenarios (such as non-natural images) to construct a broader range of reasoning data, with the goal of enhancing the model's capabilities and generalization.

---

> > > ### Author Response · Authors · 2024-11-21
> > >
> > > > ### **Ablation experiments to validate the benefits of CoM data**:
> > >
> > > Thank you for your suggestion. We have conducted ablation experiments to validate the impact of the CoM training data by training our model with and without its incorporation. The comparison results on three typical benchmark categories—TextVQA, MMVet, and MathVista—are shown in Table 4. **In conclusion, our model benefits from the CoM corpus, which contains informative visual reasoning evidence, achieving improvements of 6.6, 0.2, and 0.9 percentage points on the three benchmarks, respectively.**

---

> ### Author Response · Authors · 2024-11-25
> **Looking forward to response**
>
> Dear Reviewer sBKx,
>
> Thank you very much for investing significant effort in reviewing our paper. Regarding your concerns and questions, we have submitted point-by-point responses. We sincerely hope that our responses address your concerns adequately and we look forward to your feedback.

---

> > ### Comment · Reviewer_sBKx · 2024-11-25
> > **Good rebuttal**
> >
> > The rebuttal address most of my concerns. I will raise my score.

---

> > > ### Author Response · Authors · 2024-11-26
> > >
> > > Dear Reviewer,
> > >
> > > Thank you very much for your positive feedback.  It means a great deal to us and motivates us to strive for work that contributes more meaningfully to the community.

---

### Meta-Review · Area_Chair_fnCp · 2024-12-20

**Metareview:**

This paper introduces a mechanism called “Chain of Manipulations” inspired by human cognition, which enables VLM to solve problems step by step using evidence. With this mechanism, the model can extract unique operations and solve visual problems. In addition, it is also possible to trace the cause of errors. The paper proposes a comprehensive methodology that includes flexible operation design, an automatic data generation pipeline, a compatible VLM architecture, and a versatile training process.

The strengths of this paper are as follows. As evidence of active manipulation of visual input, the paper proposes the Chain of Manipulations (CoM), a mechanism that enables VLM to solve problems in stages. Specifically, the paper proposes a flexible data structure, an efficient data generation framework that can generate a large number of samples, a memory-based architecture that is compatible with existing VLMs, and a training process for cultivating general-purpose abilities.Furthermore, in order to utilize the expressive power of the proposed mechanism and further advance VLM research towards solving difficult mathematical problems, this paper manually annotated 7,000 high-quality mathematical graph samples with the CoM inference process.

The weaknesses of this paper are as follows. The size of the CoM dataset is limited. In order to strengthen the model's functionality and generalization, it is necessary to construct a wider range of inference data for a greater variety of scenarios. The success rate for constructing inference paths is high for common, prominent objects, but low for small, less common objects. There is a lack of comparison with research on improving the inference function of VLM.

This paper is clearly written, and all reviewers have given it positive evaluations. Based on a comprehensive judgment of the paper itself, the reviewers' comments, and the author's rebuttal, the AC believes that this paper exceeds the ICLR's acceptance threshold and that acceptance is appropriate.

**Additional Comments On Reviewer Discussion:**

Discussion of differences with previous work, limited size of the CoM dataset, ablation experiments to validate the benefit of CoM data, insignificant performance on grounding benchmarks, how CoM-train and CoM-math separately contribute to performance, etc. were discussed, the authors responded appropriately, and the reviewers were satisfied. These revisions were appropriately reflected in the paper.

---

### Decision · Program_Chairs · 2025-01-22

Accept (Poster)